# TopoDiffusionNet: A Topology-aware Diffusion Model

**Saumya Gupta, Dimitris Samaras & Chao Chen**
Department of Computer Science
Stony Brook University
Stony Brook, NY 11794, USA
{saumgupta,samaras}@cs.stonybrook.edu, chao.chen.1@stonybrook.edu

## Abstract

Diffusion models excel at creating visually impressive images but often struggle to generate images with a specified *topology*. The Betti number, which represents the number of structures in an image, is a fundamental measure in topology. Yet, diffusion models fail to satisfy even this basic constraint. This limitation restricts their utility in applications requiring exact control, like robotics and environmental modeling. To address this, we propose TopoDiffusionNet (TDN), a novel approach that enforces diffusion models to maintain the desired topology. We leverage tools from topological data analysis, particularly persistent homology, to extract the topological structures within an image. We then design a topology-based objective function to guide the denoising process, preserving intended structures while suppressing noisy ones. Our experiments across four datasets demonstrate significant improvements in topological accuracy. TDN is the first to integrate topology with diffusion models, opening new avenues of research in this area. Code available at https://github.com/Saumya-Gupta-26/TopoDiffusionNet

## 1 Introduction

Over the past few years, diffusion models have become prominent for image generation tasks (Sohl-Dickstein et al., 2015; Song & Ermon, 2019; Ho et al., 2020; Song et al., 2020a;b; Nichol & Dhariwal, 2021; Dhariwal & Nichol, 2021). Naturally, text-to-image (T2I) diffusion models are a popular choice for creating high-quality images based on textual prompts (Saharia et al., 2022a; Rombach et al., 2022; Avrahami et al., 2022; Ruiz et al., 2023; Nichol et al., 2021; Kim et al., 2022; Ramesh et al., 2021; Midjourney; OpenAI, a). Despite their ability to generate visually impressive images, T2I models are still far from the desired intelligence level. In particular, they often struggle to interpret textual prompts that involve basic reasoning and logic. This includes preserving global and semantic constraints, such as consistent number of objects as well as structural patterns (e.g., enclosed regions or loops). Improving this capability would be a step forward in the control and precision of diffusion models, moving beyond qualitative attributes such as style and texture.

Topology, in a general sense, defines how different parts of an image interact with each other, dictating their overall layout within an image. Preserving topology is essential for generating images that not only look realistic but also adhere to correct semantics. The simplest measure in topology is the Betti number, which is equivalent to the number of connected components (0-dimension) and holes/loops (1-dimension). In natural images, 0-dimensional topology corresponds to the number of distinct objects, while 1-dimensional topology refers to the number of enclosed regions. Yet, current diffusion models fail to preserve even these basic topological properties. This is particularly evident in applications such as urban planning, robotics, and environmental modeling, where it is crucial to generate scenes with a specific topology or number of entities (like animals or road intersections). Fig. 1(a-b) shows instances where popular T2I diffusion models fail to generate images with the specified topology, such as specific numbers of animals, or holes/regions in road layouts.

Recognizing the limited spatial reasoning of T2I models, existing methods use *spatial maps* (such as object masks, edge maps, etc) to control the generated images (Zhang et al., 2023; Bar-Tal et al.,

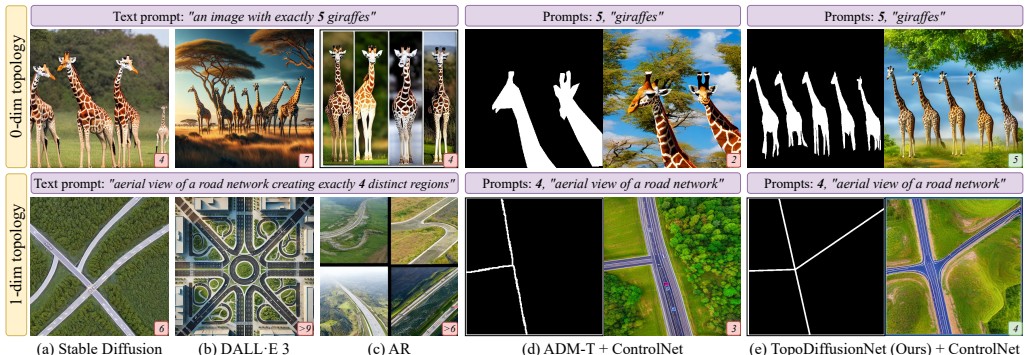

Figure 1: **Comparison of existing diffusion models in preserving topological constraints.** Top row: 0-dim topological constraint to generate exactly five giraffes. Bottom row: 1-dim topological constraint to generate a road layout with exactly four distinct regions. Text-to-image methods like (a) Stable Diffusion (Rombach et al., 2022) and (b) DALL·E 3 (OpenAI, a) struggle to respect both 0-dim and 1-dim constraints. (c) Attention Refocusing (AR) (Phung et al., 2024) requires bounding boxes for each object but struggles with higher object counts and often creates partitioned images. (d)-(e) shows a two-step process: mask generation followed by ControlNet (Zhang et al., 2023) rendering. (d) ADM-T generates masks by fine-tuning ADM (Dhariwal & Nichol, 2021) with the topological constraint as a condition, but this alone is insufficient. (e) Our TopoDiffusionNet, trained with a topology-based objective function, generates masks with the precise number of objects or regions, which when fed to ControlNet generates the desired image of five giraffes (top row) and four regions (bottom row). Giraffe/region counts are noted in the bottom-right inset of each image.

2023; Bashkirova et al., 2023; Huang et al., 2023; Mou et al., 2023; Li et al., 2023). These methods have indeed shown promise, offering a more guided approach to image generation that aligns closely with the provided controls. Nonetheless, generating the spatial map itself is a bottleneck: an automatic way to generate spatial maps with a desired topology is an unaddressed problem.

In this work, we address the challenge of generating topologically faithful images using diffusion models. Our focus is on generating images with a specific *topology*, defined by properties such as the number of connected components (0-dimension) or loops/holes (1-dimension). These topological structures, quantified by Betti numbers, serve as our topological constraints. Spatial maps such as masks have shown success in guiding the semantics of the generated image. Leveraging this, we propose using diffusion models to automate generating masks that satisfy the desired topological constraint. A straightforward approach is to condition the diffusion model on the constraint information and fine-tune it on enough samples. However, as shown in Fig. 1(d), we find that conditioning on the constraint alone falls short of effectively preserving the topology of the generated masks.

We propose TopoDiffusionNet (TDN), a novel approach that incorporates topology to guide the mask generation process. To ensure the final mask satisfies the topological constraint, a topology-aware objective function is necessary for steering the denoising process so that each timestep is one step closer to preserving the desired topological constraint. However, designing such a function is not straightforward. The intermediate timesteps are very noisy – especially at larger timesteps – so extracting meaningful information from them is challenging. We thus need tools that are robust to noise. This leads us to *persistent homology* (Edelsbrunner et al., 2002; Edelsbrunner & Harer, 2010), a mathematical theory that can, amidst the noise, extract the topological structures within an image. Using persistent homology, we can partition an image in terms of topology: separating out the significant structures from the noisy ones. We can thus design a dedicated objective function to preserve the significant structures and suppress the rest. The function guides the denoising process to progress in such a way so as to ultimately preserve the topology at the final timestep, as we see in Fig. 1(e). In summary, our contributions are as follows:

- To the best of our knowledge, we are the first to integrate topology with diffusion models to address topologically faithful image generation in both 0-dimension and 1-dimension. Specifically, we generate spatial maps (masks) to tackle the challenge of generating an image with a specific number of structures.

- We present TopoDiffusionNet (TDN), which utilizes a topology-based objective function to improve diffusion models' ability to follow simple topological constraints. It serves as a denoising loss, guiding the diffusion denoising process in a topology-aware manner.
- We evaluate TDN on four datasets to demonstrate its versatility and effectiveness. TDN exhibits large improvements in maintaining the topological integrity of the generated image.

The success of our method suggests a surprising harmony between diffusion models and topology. Diffusion models are trained to denoise but are rather hard to control for preserving global semantics. Meanwhile, topological methods such as persistent homology provide a principled solution to extract global structural information from a noisy input, and can thus successfully guide the diffusion model. We hope the coupling of diffusion models and topology, as well as the techniques developed in this paper, will shed light on more sophisticated control of these generative models in the near future.

## 2 RELATED WORK

**Diffusion models.** Diffusion models, first introduced by Sohl-Dickstein et al. (2015), are now prevalent in image generation (Ho et al., 2020; Song et al., 2020a;b; Nichol & Dhariwal, 2021; Dhariwal & Nichol, 2021), evolving from unconditional models, to conditional models using class labels (Ho & Salimans, 2022; Dhariwal & Nichol, 2021), and later to text-to-image models (Saharia et al., 2022a; Rombach et al., 2022; Avrahami et al., 2022; Ruiz et al., 2023; Nichol et al., 2021; Kim et al., 2022; Ramesh et al., 2021; Midjourney; OpenAI, a). However, textual prompts have limitations in controlling spatial composition, like layouts and poses. Recognizing this, several works propose to use spatial maps (such as masks, edge maps, etc) as condition to guide the image generation process (Zhang et al., 2023; Qin et al., 2023; Zhao et al., 2024; Bar-Tal et al., 2023; Bashkirova et al., 2023; Huang et al., 2023; Mou et al., 2023). These approaches aim to overcome the limitations of text-based conditioning by providing more explicit spatial guidance.

**Numeric control in diffusion models.** Betti numbers quantify topological structures, such as the number of connected components (0-dimension) or loops/holes (1-dimension), which is conceptually related to the task of counting. Recent works have explored approaches to enhance the counting performance in diffusion models. Paiss et al. (2023) enhances CLIP's (Radford et al., 2021) text embeddings for counting-aware text-to-image (T2I) generation via Imagen (Saharia et al., 2022b). While this results in some improvement, the performance is still limited, supporting our motivation that T2I models often struggle with textual prompts involving semantic reasoning. Layout-based methods (Chen et al., 2024; Phung et al., 2024; Farshad et al., 2023) use layout maps, that is, maps containing bounding boxes of each object/entity, to guide the reverse diffusion process. While these methods show promise, they are not scalable as their complexity increases with the number of objects. Furthermore, focusing on these boxes often results in images that appear partitioned, as shown in Fig. 1(c) top row, where the backgrounds of each giraffe differ. Finally, all of the methods mentioned above are limited to 0-dimensional topological structures (i.e., discrete objects) and do not extend to higher-dimensional topological constraints.

**Deep learning with topology.** Methods from algebraic topology, under the name of *topological data analysis* (TDA) (Carlsson, 2009), have found use in various machine learning problems owing to their versatility (handling data such as images, time-series, graphs, etc.) and robustness to noise. The most widely-used tool from TDA, *persistent homology* (PH) (Edelsbrunner et al., 2002; Edelsbrunner & Harer, 2010), has been applied to several image classification (Peng et al., 2024; Wang et al., 2021; Du et al., 2022; Hofer et al., 2017; Chen et al., 2019) and segmentation tasks (Abousamra et al., 2021; Clough et al., 2019; Hu et al., 2019; Clough et al., 2020; Stucki et al., 2023; Byrne et al., 2022; He et al., 2023; Xu et al., 2024) as it can track topological changes at multiple intensity values. Other TDA theories like discrete Morse theory (Dey et al., 2019; Hu et al., 2021; 2023; Gupta et al., 2024; Banerjee et al., 2020), topological interactions (Gupta et al., 2022), and center-line transforms (Shit et al., 2021; Wang et al., 2022a), have also enhanced performance in these areas.

In the realm of generative models, TDA has been used with generative adversarial networks (GANs) (Goodfellow et al., 2014) to evaluate performance through topology comparisons (Khrulkov & Oseledets, 2018), and enhance image quality via topological priors (Brüel-Gabrielsson et al., 2019) and PH-based loss functions (Wang et al., 2020). These methods have mainly focused on quality enhancement without directly controlling the topological characteristics of the images. In the case of diffusion models, incorporating TDA has not yet been explored. Our work, TopoDif-

fusionNet, represents a unique effort in this direction, employing PH within diffusion models to control the topology of the generated images. This is a significant shift from existing applications of TDA in generative models, moving beyond quality improvement to precise topological control.

## 3 METHODOLOGY

Given a topological constraint $c$, our goal is to generate a *mask* containing $c$ number of *structures*. A structure can either correspond to an object or a hole/region. We illustrate these structures in Fig. 2. In (a), the four objects are in white. In (b), the four holes correspond to the four black regions/partitions the white lines create with the border. In the formal language of algebraic topology (Munkres, 2018), $c$ is the Betti number, that is, it is the rank of the homology group, in which objects (connected components) correspond to the 0-dimensional (0-dim) homology classes and holes/loops/regions correspond to the 1-dimensional (1-dim) homology classes.

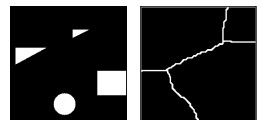

(a) Four objects (b) Four holes

Figure 2: Illustration of topological structures.

To enforce the topological constraint $c$, the diffusion model needs to be conditioned on it. However, $c$ alone is not sufficient to control the topology of the final generated image[1]. Therefore, we propose a topology-based objective function $\mathcal{L}_{\text{top}}$ to guide the reverse denoising diffusion process at each timestep during training. $\mathcal{L}_{\text{top}}$ uses persistent homology (Edelsbrunner et al., 2002; Edelsbrunner & Harer, 2010) to distinguish between the desired and the spurious structures, aiming to enhance the former and reduce the latter to ensure the topology matches $c$ closely. Fig. 3 provides an overview.

The rest of this section is organized as follows. We briefly discuss diffusion models in Sec. 3.1, followed by a quick background on persistent homology in Sec. 3.2. We tie these concepts together to finally introduce our method TopoDiffusionNet (TDN) in Sec. 3.3.

### 3.1 DIFFUSION MODELS

Diffusion models (Ho et al., 2020) are able to sample images from the training data distribution $p(x_0)$ by iteratively denoising random Gaussian noise in $T$ timesteps. The framework consists of a forward and a reverse process.

In the forward process, at every timestep $t \in T$, Gaussian noise is added to the clean image $x_0 \sim p(x_0)$ until the image becomes an isotropic Gaussian. The forward noising process is denoted by $q(x_t \mid x_0) \coloneqq \mathcal{N}(x_t; \sqrt{\bar{\alpha}_t}x_0, (1 - \bar{\alpha}_t)\boldsymbol{I})$, which can be rewritten as,

$$x_t = \sqrt{\bar{\alpha}_t}x_0 + \sqrt{(1 - \bar{\alpha}_t)}\epsilon \tag{1}$$

$\epsilon \sim \mathcal{N}(\boldsymbol{0}, \boldsymbol{I})$ is a noise variable, $\bar{\alpha}_t$ is the noise scale at timestep $t$, and $\mathcal{N}$ is the normal distribution.

The reverse process aims to learn the posterior distribution $q(x_{t-1} \mid x_t, x_0)$, using which we can recover $x_{t-1}$ given $x_t$. This is typically done by training a denoising neural network (U-Net (Ronneberger et al., 2015)) with network parameters $\theta$. The denoising model $\epsilon_\theta(x_t, t)$ takes the noisy input $x_t$ at timestep $t$ and predicts the noise $\epsilon$ added in Eq. (1) of the forward process. The model is trained using the simplified objective function $\mathcal{L}_{\text{simple}}$ (Ho et al., 2020): $\mathcal{L}_{\text{simple}} = \mathbb{E}_{t,x_0,\epsilon}\left[\|\epsilon_\theta(x_t, t) - \epsilon\|_2^2\right]$. In our case, we additionally provide the topological constraint $c$ as a condition to control the topology of the generated image. Thus, the denoising model becomes $\epsilon_\theta(x_t, c, t)$, where $c$ is injected into the denoising neural network.

During training, although the denoising model predicts the noise $\epsilon_\theta(x_t, c, t)$ at timestep $t$, we can deterministically (without iterative sampling) recover the predicted noiseless image $\hat{x}_0^t$ (an estimate of the true $x_0$) from Eq. (1) as,

$$\hat{x}_0^t = \frac{1}{\sqrt{\bar{\alpha}_t}}\left(x_t - \sqrt{1 - \bar{\alpha}_t}\epsilon_\theta(x_t, c, t)\right) \tag{2}$$

This alternate form of the prediction will enable us to compute $\mathcal{L}_{\text{top}}$ as we will see in Sec. 3.3.

### 3.2 PERSISTENT HOMOLOGY

Persistent homology (PH) (Edelsbrunner et al., 2002; Edelsbrunner & Harer, 2010), owing to its differentiable nature, is an attractive candidate for integrating topological information into the training of deep learning methods. In the case of image data, it can detect the changes in topological

---

[1]In this section, we use 'image' to mean the mask.

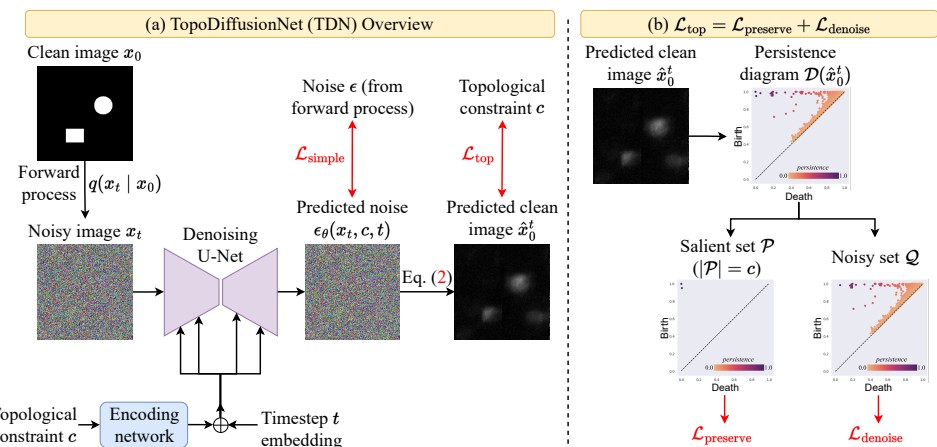

Figure 3: (a) TDN overview: We condition the diffusion model on the topological constraint $c$ (here $c = 2$). During training, we first add noise $\epsilon$ to the input $x_0$ using the forward process (Eq. (1)) to obtain $x_t$, where $t$ is sampled uniformly. The U-Net is trained as part of the reverse process, predicting the added noise $\epsilon_\theta(x_t, c, t)$, with which we obtain the noiseless image $\hat{x}_0^t$ (Eq. (2)). Alongside the standard loss $\mathcal{L}_{\text{simple}}$, we propose $\mathcal{L}_{\text{top}}$ to enforce the topological integrity of $\hat{x}_0^t$. (b) To compute $\mathcal{L}_{\text{top}}$, the persistence diagram $\mathcal{D}(\hat{x}_0^t)$ captures all the topological structures in $\hat{x}_0^t$, partitioning them into salient/desired structures $\mathcal{P}$ and noisy ones $\mathcal{Q}$. Terms $\mathcal{L}_{\text{preserve}}$ and $\mathcal{L}_{\text{denoise}}$ respectively amplify $\mathcal{P}$ and suppress $\mathcal{Q}$, guiding the training to eventually satisfy $c$.

structures (connected components and holes) across a varying threshold (also called the filtration value). More importantly, persistent homology is robust to noise, that is, it can extract these structures even in noisy scenarios. Structures that exist for a wide range of thresholds are *salient*, while the remaining structures are deemed as *noise* in the image.

In our setting, during training, we can apply persistent homology to every intermediate image $\hat{x}_0^t$ (from Eq. (2)) predicted by the diffusion model at timestep $t$. For ease of reference, we denote $\hat{x}_0^t$ by $I$, having size $h \times w$. In practice, $I$ has continuous probability values in a normalized range, say, $[0, 1]^2$. We now consider *super-level sets* of $I$, i.e., the set of pixels $(i, j)$ for which $I_{ij}$ is above some threshold value $u$. Let $\mathcal{S}$ denote the super-level set, then, $\mathcal{S}(u) := \{(i, j) \in [1, h] \times [1, w] \mid I_{ij} \geq u\}$. This is nothing but thresholding, and we call the resulting binary image the super-level set at $u$, $S(u)$. Decreasing $u$ continuously generates a sequence of sets, i.e. a filtration, which grows as the threshold parameter $u$ is brought down: $\varnothing \subseteq \mathcal{S}(1) \subseteq \mathcal{S}(u_1) \subseteq \mathcal{S}(u_2) \subseteq \cdots \subseteq \mathcal{S}(0) = [1, h] \times [1, w]$. We demonstrate this filtration in Fig. 4.

When $u$ is high, only a few pixels can exceed the threshold, and hence the size of $\mathcal{S}(1)$ is small (an almost black image). As $u$ decreases, new pixels join the set, and topological structures in $\mathcal{S}$ are created and destroyed. Eventually, at $u = 0$, the entire image is in the super-level set. In this manner, persistent homology can track the evolution of all the topological structures.

The output of the persistent homology algorithm includes the *birth* and *death* threshold values for each topological structure. We can keep track of the birth (creation) $b$ and death (destruction) $d$ thresholds of all the topological structures and put the tuples $(b, d)$ in a diagram – the *persistence diagram* $\mathcal{D}$ – where the $y$-axis represents birth and the $x$-axis death.[3] The persistence diagram is thus a graphical representation of topological structures throughout the filtration process, consisting of multiple dots in a 2-dimensional plane (see Fig. 4). These dots are called persistent dots, where each dot corresponds to one topological structure. The *persistence*, or the lifetime of a structure, is given by the difference between its death and birth times. Structures that persist over a wide range of thresholds are considered significant or salient, indicating stable and prominent structures within the image, while short-lived structures are the noise. The diagonal $b = d$ represents structures of zero persistence and dots far from the diagonal represent salient structures with high persistence.

---

[2]The range is typically $[-1, 1]$ in implementation.

[3]If using a sub-level instead of super-level set filtration, the diagram would have the $x$-axis as birth, and the $y$-axis as death.

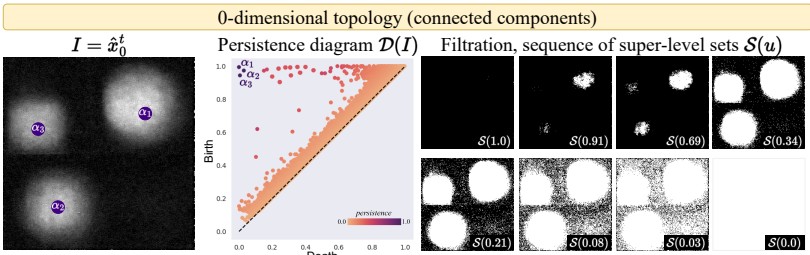

Figure 4: Illustration of persistent homology and persistence diagrams of 0-dim topological structures (connected components). Despite the noise, we can visually see three prominent structures $\alpha_1, \alpha_2, \alpha_3$ in $I$. In the topological space, $\alpha_1, \alpha_2, \alpha_3$ thus appear in the top-left corner of the persistence diagram $\mathcal{D}(I)$, persisting through most of the filtration $\mathcal{S}$. All the remaining connected components are noisy, persisting over a short threshold in $\mathcal{S}$, thus appearing closer to the diagonal in $\mathcal{D}(I)$. Persistence diagrams are useful to distinguish between salient and noisy structures in an image. The equivalent illustration for 1-dim topological structures (holes) is in Appendix A.

With this information, we can identify noisy structures due to their low persistence and proximity to the diagonal, allowing us to filter them out and retain the salient topological structures. The number of salient structures we retain is precisely the number of structures we desire in the final image. As we show in the next subsection, we compute the persistence diagram of the predicted image $\hat{x}_0^t$ to optimize it from a topological perspective.

### 3.3 PROPOSED TOPODIFFUSIONNET (TDN)

**Conditioning.** We condition the diffusion model on $c$ to enable it to generate a mask containing exactly $c$ number of structures. We follow Nichol & Dhariwal (2021) to inject the condition information. We first obtain an embedding of $c$ from a trainable network composed of a few linear layers. Next, we inject the embedding through the same pathway as the timestep embedding of $t$. Consequently, both embeddings are passed to residual blocks throughout the denoising model.

**Objective function $\mathcal{L}_{\text{top}}$.** Conditioning alone is not sufficient to control the topology of the generated image. To address this, we introduce $\mathcal{L}_{\text{top}}$, a topology-based objective function to force the predicted image at every timestep to preserve $c$ as closely as possible. Since the diffusion model is parameterized to predict in the noise space, directly analyzing topology from this noise is not meaningful. We need to map the prediction from the noise space back to the image space in order to infer the topology. From Eq. (2), we obtain $\hat{x}_0^t$ – an estimate of the noiseless image $x_0$ at timestep $t$ – from $\epsilon_\theta(x_t, c, t)$. The estimate $\hat{x}_0^t$ is noisy, especially when $t$ is large, making it ideal to use the theory of persistent homology to separate out its salient structures from the noisy ones.

Given the prediction $\hat{x}_0^t$, we compute its persistence diagram $\mathcal{D}(\hat{x}_0^t)$ containing either 0-dim or 1-dim information based on the desired topological structure (object or regions). Recall that we desire $c$ topological structures in the predicted image. For a persistent dot $p \in \mathcal{D}$, with birth $b_p$ and death $d_p$, its persistence value, $|b_p - d_p|$, measures its significance, according to the theory. We rank all dots in $\mathcal{D}$ by their persistence values in descending order. The top $c$ dots are the structures we aim to preserve, reflecting our desired topology, whereas the rest denote noisy structures to be suppressed/denoised. Thus, we decompose the diagram $\mathcal{D}$ into two disjoint sets, $\mathcal{D} = \mathcal{P} \bigcup \mathcal{Q}$, where $\mathcal{P}$ contains the $c$ largest persistence dots ($|\mathcal{P}| = c$), while $\mathcal{Q}$ contains all the remaining dots.

To constrain $\hat{x}_0^t$ to have $c$ structures in the *image space*, in the *topological space* we need to maximize the persistence or saliency of the dots $p \in \mathcal{P}$, and suppress all the noisy dots $p \in \mathcal{Q}$. To achieve this, we introduce two loss terms:

$$\mathcal{L}_{\text{preserve}} = -\sum_{p \in \mathcal{P}} |b_p - d_p|^2 \quad \text{and} \quad \mathcal{L}_{\text{denoise}} = \sum_{p \in \mathcal{Q}} |b_p - d_p|^2 \tag{3}$$

$$\mathcal{L}_{\text{top}} = \mathcal{L}_{\text{preserve}} + \mathcal{L}_{\text{denoise}} \tag{4}$$

Minimizing $\mathcal{L}_{\text{top}}$ is equivalent to maximizing the saliency of the top $c$ structures (via $\mathcal{L}_{\text{preserve}}$) and suppressing the saliency of the rest (via $\mathcal{L}_{\text{denoise}}$). In the ideal case, $\mathcal{L}_{\text{top}} = -c$, as each of the top $c$ structures will have a persistence of 1, while all the noisy structures will have zero persistence.

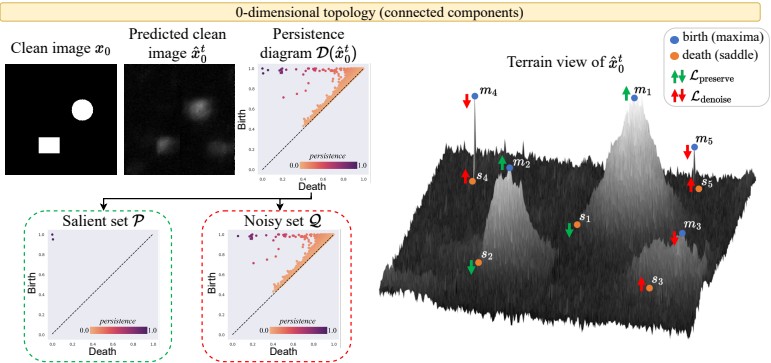

Figure 5: Illustration of $\mathcal{L}_{\text{preserve}}$ and $\mathcal{L}_{\text{denoise}}$ for 0-dim connected components, with $c = 2$ as seen in $x_0$. After computing $\mathcal{D}(\hat{x}_0^t)$, we partition it into sets $\mathcal{P}$ (the top $c$ structures) and $\mathcal{Q}$ (remaining ones). For each dot $p \in \mathcal{D}(\hat{x}_0^t)$, the birth and death values respectively correspond to local maxima $m_p$ and saddles $s_p$ in $\hat{x}_0^t$. In the terrain view of $\hat{x}_0^t$, structures $(m_1, s_1)$ and $(m_2, s_2)$ belong to $\mathcal{P}$; hence optimizing $\mathcal{L}_{\text{preserve}}$ increases their saliency by increasing $\hat{x}_0^t(m_1), \hat{x}_0^t(m_2)$ and decreasing $\hat{x}_0^t(s_1), \hat{x}_0^t(s_2)$. All the remaining $n$ structures $(m_3, s_3), (m_4, s_4), \cdots, (m_n, s_n)$ belong to $\mathcal{Q}$. Optimizing $\mathcal{L}_{\text{denoise}}$ suppresses these noisy structures by decreasing $\hat{x}_0^t(m_3), \hat{x}_0^t(m_4), \cdots, \hat{x}_0^t(m_n)$ and increasing $\hat{x}_0^t(s_3), \hat{x}_0^t(s_4), \cdots, \hat{x}_0^t(s_n)$. The equivalent figure for 1-dim holes is in Appendix B.

The image $\hat{x}_0^t$ will then have exactly $c$ topological structures as desired. In the absence of $\mathcal{L}_{\text{top}}$, if the denoising process were to originally result in $> c$ structures, now $\mathcal{L}_{\text{denoise}}$ will suppress all the extra/noisy structures, preventing them from appearing in the final clean image. One concern is whether there could be less than $c$ structures in total. In practice, at large timesteps $t$, $\hat{x}_0^t$ always has several thousand noisy topological structures. Thus, if the denoising process were to originally proceed with $< c$ structures, $\mathcal{L}_{\text{preserve}}$ will now increase the persistence of a less salient dot to ensure the final image has exactly $c$ structures.

**Implementation and differentiability.** For every topological structure, the birth and death values $b$ and $d$ correspond to function values of a maximum-saddle pair; $b$ and $d$ are function values of a local maximum $m$ and a saddle point $s$, respectively. These pairs can be determined by the almost linear union-find algorithm (Edelsbrunner & Harer, 2010; Ni et al., 2017) which locates and pairs all local maxima and saddle points to reflect the birth and death of topological structures. For every persistent dot $p \in \mathcal{D}$, let $m_p$ and $s_p$ respectively denote the 2D coordinates of the corresponding local maximum and saddle point in the prediction $\hat{x}_0^t$. Then, Eq. (3) and Eq. (4) can be rewritten as,

$$\mathcal{L}_{\text{top}} = -\sum_{p \in \mathcal{P}} |\hat{x}_0^t(m_p) - \hat{x}_0^t(s_p)|^2 + \sum_{p \in \mathcal{Q}} |\hat{x}_0^t(m_p) - \hat{x}_0^t(s_p)|^2 \tag{5}$$

We illustrate this in Fig. 5. During training, for every topological structure to preserve, i.e., $p \in \mathcal{P}$, the function $\mathcal{L}_{\text{preserve}}$ increases the intensity value $\hat{x}_0^t(m_p)$ at the local maximum $m_p$ and decreases $\hat{x}_0^t(s_p)$ at the saddle point $s_p$. This strengthens the saliency of the desired structures. At the same time, to prevent exceeding $c$ structures in the final image, $\mathcal{L}_{\text{denoise}}$ suppresses structures $p \in \mathcal{Q}$ by reducing the intensity value $\hat{x}_0^t(m_p)$ at the local maximum $m_p$ while increasing $\hat{x}_0^t(s_p)$ at the saddle point $s_p$. $\mathcal{L}_{\text{denoise}}$ forces $\hat{x}_0^t(m_p)$ to be equal to $\hat{x}_0^t(s_p)$, leading to a homogenous region. This effectively eliminates the noisy structure, as it was neither born nor died, rendering it non-existent.

With this, we see that $\mathcal{L}_{\text{top}}$ is differentiable, as Eq. (5) is written as polynomials of the prediction $\hat{x}_0^t$ at certain pixels. From Eq. (5) we can compute the gradient of $\mathcal{L}_{\text{top}}$ with respect to $\hat{x}_0^t$, and via chain rule, we can ultimately compute the gradient with respect to the denoising model's parameters $\theta$. The training optimization adjusts $\theta$ to ensure that the topological space, i.e. the persistence diagram $\mathcal{D}(\hat{x}_0^t)$, has exactly $c$ persistent dots, in turn resulting in $c$ structures in the image space $\hat{x}_0^t$.

**End-to-end training.** The overall training objective $\mathcal{L}_{\text{total}}$ of TDN is formulated as: $\mathcal{L}_{\text{total}} = \mathcal{L}_{\text{simple}} + \lambda \mathcal{L}_{\text{top}}$ where $\lambda$ is the loss weight. The standard denoising loss $\mathcal{L}_{\text{simple}}$ produces visually good results, whereas $\mathcal{L}_{\text{top}}$ helps respect the topological constraint $c$.

## 4 EXPERIMENTS

**Datasets.** We train ADM-T and TDN on four datasets: **Shapes**, **COCO** (Caesar et al., 2018), **CREMI** (Funke et al., 2016), and **Google Maps** (Isola et al., 2017). The Shapes dataset is a synthetic dataset created by us that contains objects such as circles, triangles, and/or rectangles. For CREMI, COCO, and Google Maps, we train the diffusion models on their segmentation masks. For COCO, we select masks which contain at least one instance of the super category 'animal'. For COCO and Shapes, we use 0-dim, the number of connected components, as the topological constraint. For COCO, we also use the animal class as a condition to generate masks of specific animals. CREMI is an Electron Microscopy dataset, and Google Maps contains aerial photos from New York City. For CREMI and Google Maps, we use 1-dim, the number of holes, as the topological constraint. Each of the datasets contains masks consisting of up to ten structures. More details are in Appendix C.

**Baselines.** Stable Diffusion (Rombach et al., 2022) and DALL·E 3 (OpenAI, a) are popular T2I diffusion models. Attention Refocusing (AR) (Phung et al., 2024) uses layout maps (bounding boxes for each object) generated by GPT-4 (Achiam et al., 2023) to guide the reverse process.

**Implementation details.** Our work extends the ADM (Dhariwal & Nichol, 2021) diffusion model. We use 'ADM-T' to denote the modification of using a topological constraint as a condition. We obtain an embedding of the constraint using an encoding network. Following the approach in Nichol & Dhariwal (2021), we then feed this embedding to all the residual blocks in the network by adding it to the timestep embedding. For COCO, we similarly inject animal class embedding to further control the generated mask. For every dataset, we use $256 \times 256$ as the image resolution. Our diffusion models use a cosine noise scheduler (Nichol & Dhariwal, 2021), with $T = 1000$ timesteps for training. During inference, however, we use only 50 steps of DDIM (Song et al., 2020a) sampling. For the ADM-T baseline, we load a pretrained checkpoint (OpenAI, b) and then fine-tune on our datasets using the constraint information as condition. For TDN, we follow the same approach but additionally use $\mathcal{L}_{\text{top}}$ in the training. To compute persistent homology, we use the Cubical Ripser (Kaji et al., 2020) library. More details are listed in Appendix D.

**Evaluation metrics.** To evaluate whether the generated image satisfies the input constraint, we use metrics such as Accuracy, Precision, and F1. We report the mean and standard deviation of the results across different constraint values. To measure the performance, for 0-dim, we generate 50 samples per constraint $c \in [1, 10]$ per animal/shape category (resulting in 5K images for COCO). We similarly generate 50 images per constraint $c \in [1, 10]$ for the 1-dim datasets. We perform the unpaired t-test (Student, 1908) (95% confidence interval) to determine the statistical significance of the improvement. In all the tables, performances that are statistically significantly better are in **bold**.

### 4.1 RESULTS

**Qualitative and quantitative results for 0-dim.** In Fig. 6 and Tab. 1, we present qualitative and quantitative comparisons, respectively, of pretrained Stable Diffusion, DALL·E 3, AR, ADM-T, and our proposed TDN. In Appendix E, we provide constraint-wise results. As ADM-T and TDN produce masks, we employ pretrained ControlNet (Zhang et al., 2023; Lvmin Zhang) (SD1.5 backbone) to create textured images from these masks. In practice, any of the methods (Qin et al., 2023; Zhao et al., 2024; Bar-Tal et al., 2023; Bashkirova et al., 2023; Huang et al., 2023; Mou et al., 2023) could also be used for this purpose, but we chose ControlNet for its simplicity. From Fig. 6, we see that T2I models Stable Diffusion and DALL·E 3 are unable to respect the explicit topological constraint of generating $c$ objects. Due to limited spatial and semantic reasoning, both methods have the lowest performance in Tab. 1. AR uses layouts generated by GPT-4, which improves over T2I models, as seen in Tab. 1. In general, however, GPT-4 does not have much spatial awareness and when the count is high, the bounding boxes tend to either be too small or highly overlap with each other. This leads to incorrect counts in the generated image as seen in Fig. 6. Additionally, when the object count is high, AR often produces fragmented results, with objects isolated within assigned areas, leading to a divided and visually disjointed image. ADM-T, despite being conditioned on the constraint $c$, also falls short of satisfying the constraint. This indicates that the constraint alone is not powerful enough to influence the global reasoning of the model. In contrast, TDN preserves the constraint better than ADM-T, as evident from the quantitative results in Tab. 1. TDN demonstrates significant improvements across all metrics. $\mathcal{L}_{\text{preserve}}$ aims to retain at least $c$ objects, while $\mathcal{L}_{\text{denoise}}$ aims to maintain atmost $c$ objects. Thus optimizing both simultaneously via training with $\mathcal{L}_{\text{top}}$ helps the diffusion model to preserve $c$ objects in the generated mask.

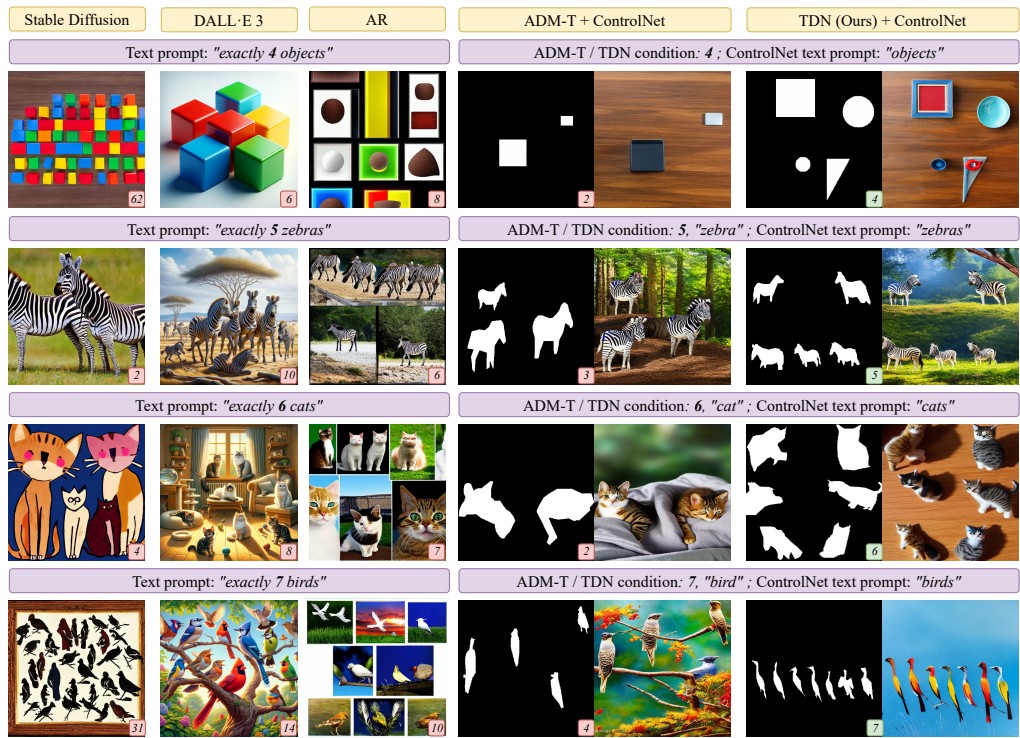

Figure 6: Qualitative results for 0-dim topological constraint. Row 1: Shapes dataset; Rows 2-4: COCO dataset. ADM-T and TDN take the constraint as the condition (purple box), and also the animal class for COCO. Stable Diffusion, DALL·E 3, and AR take the equivalent text prompt as input. Object/animal counts are noted in the bottom-right inset of each image/mask.

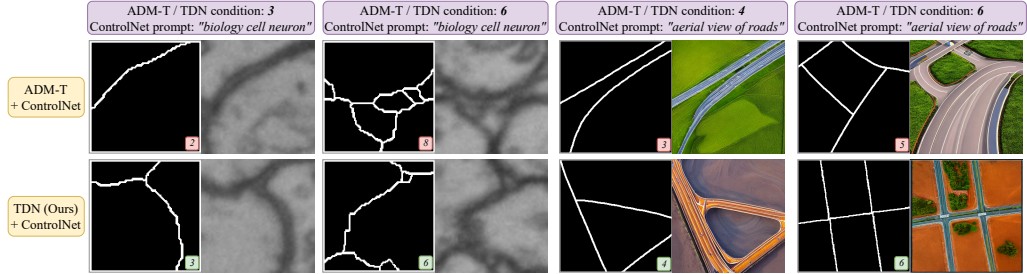

Figure 7: Qualitative results for the 1-dim topological constraint. ADM-T and TDN take the constraint (in the purple box) as the condition. Columns 1-2: CREMI. Columns 3-4: Google Maps. Number of holes within each mask is noted in its bottom-right inset.

**Qualitative and quantitative results for 1-dim.** In Fig. 7, we provide the qualitative comparison for ADM-T and TDN. We exclude results from Stable Diffusion, DALL·E 3, and AR as these methods are limited to generating distinct objects (0-dim topology) and currently cannot handle generating distinct holes/regions (1-dim topology). As holes are a complex topological constraint, they are challenging to describe in words, and hence the performance of such methods is limited (see Fig. 1 and Appendix G). Similar to the 0-dim case, ADM-T struggles with the topological constraint. The quantitative results in Tab. 1 highlight that preserving 1-dim topology, which involves boundaries spanning across the mask, is more challenging than 0-dim topology. This complexity necessitates powerful global reasoning capabilities from the diffusion model, an area where ADM-T shows limited performance. However, with $\mathcal{L}_{\text{top}}$, TDN achieves substantial improvements across all metrics, with $\mathcal{L}_{\text{preserve}}$ and $\mathcal{L}_{\text{denoise}}$ both working to retain exactly $c$ holes in the generated mask.

**Additional comparison.** Paiss et al. (2023) enhances CLIP's text embeddings for counting and uses them to show counting-aware T2I generation via Imagen. We report the Accuracy of our TDN results in their setting in Tab. 2, showing significant performance improvement. This supports our motivation that T2I models often struggle with textual prompts involving semantic reasoning.

Table 1: Quantitative comparison on preserving the topological constraint $c$

| Dataset | Method | Accuracy ↑ | Precision ↑ | F1 ↑ |
|---|---|---|---|---|
| **Shapes** | Stable Diffusion (Runway) | $0.6381 \pm 0.2559$ | $0.6660 \pm 0.1759$ | $0.6537 \pm 0.2198$ |
| | DALL·E 3 (OpenAI, a) | $0.6857 \pm 0.2561$ | $0.7059 \pm 0.3198$ | $0.6956 \pm 0.2649$ |
| | AR (Phung et al., 2024) | $0.7384 \pm 0.2178$ | $0.7596 \pm 0.2039$ | $0.7474 \pm 0.2165$ |
| | ADM-T | $0.7500 \pm 0.1889$ | $0.7809 \pm 0.1582$ | $0.7651 \pm 0.1210$ |
| | TDN (Ours) | $\mathbf{0.9478 \pm 0.0420}$ | $\mathbf{0.9499 \pm 0.0492}$ | $\mathbf{0.9488 \pm 0.0370}$ |
| **COCO (Animals)** | Stable Diffusion (Runway) | $0.4686 \pm 0.2361$ | $0.5154 \pm 0.1747$ | $0.4909 \pm 0.1976$ |
| | DALL·E 3 (OpenAI, a) | $0.5162 \pm 0.3674$ | $0.5491 \pm 0.2724$ | $0.5194 \pm 0.3256$ |
| | AR (Phung et al., 2024) | $0.6379 \pm 0.2062$ | $0.7360 \pm 0.1658$ | $0.6611 \pm 0.1851$ |
| | ADM-T | $0.6685 \pm 0.1485$ | $0.6917 \pm 0.1079$ | $0.6799 \pm 0.1931$ |
| | TDN (Ours) | $\mathbf{0.8557 \pm 0.0805}$ | $\mathbf{0.8670 \pm 0.0636}$ | $\mathbf{0.8613 \pm 0.0970}$ |
| **Google Maps** | ADM-T | $0.5494 \pm 0.1386$ | $0.5642 \pm 0.1861$ | $0.5567 \pm 0.1185$ |
| | TDN (Ours) | $\mathbf{0.8318 \pm 0.1159}$ | $\mathbf{0.8471 \pm 0.1797}$ | $\mathbf{0.8394 \pm 0.1969}$ |
| **CREMI** | ADM-T | $0.5357 \pm 0.1879$ | $0.4777 \pm 0.1797$ | $0.4881 \pm 0.1571$ |
| | TDN (Ours) | $\mathbf{0.7785 \pm 0.1901}$ | $\mathbf{0.8142 \pm 0.1925}$ | $\mathbf{0.7959 \pm 0.1659}$ |

Table 2: Additional baseline

| Method | Accuracy ↑ |
|---|---|
| Paiss et al. (2023) | 0.5018 |
| TDN (Ours) | **0.7969** |

Table 3: Ablation on loss terms

| $\mathcal{L}_{\text{preserve}}$ | $\mathcal{L}_{\text{denoise}}$ | Accuracy ↑ |
|---|---|---|
| ✗ | ✗ | $0.7500 \pm 0.1889$ |
| ✓ | ✗ | $0.8926 \pm 0.1821$ |
| ✗ | ✓ | $0.9186 \pm 0.1129$ |
| ✓ | ✓ | $\mathbf{0.9478 \pm 0.0420}$ |

Table 4: Ablation on $\lambda$

| Loss weight $\lambda$ | Accuracy ↑ |
|---|---|
| 0 | $0.7500 \pm 0.1889$ |
| 1e-3 | $0.9066 \pm 0.0612$ |
| 1e-5 | $\mathbf{0.9478 \pm 0.0420}$ |
| 1e-7 | $0.9176 \pm 0.0407$ |
| Min-SNR (5) | $0.9286 \pm 0.0320$ |

**Effect across timesteps.** While we use 50 steps of DDIM (Song et al., 2020a) sampling for inference, the training was conducted with $T = 1000$ timesteps using DDPM (Ho et al., 2020). To understand the effect of $\mathcal{L}_{\text{top}}$ throughout training, we analyze intermediate results from the DDPM inference procedure. In Fig. 8, we visualize $\hat{x}_0^t$ across different timesteps on the Shapes dataset, highlighting the impact of $\mathcal{L}_{\text{top}}$ on denoising efficiency. Notably, TDN achieves a closer approximation to the true $x_0$ by timestep 750, nearly 200 timesteps before ADM-T. Furthermore, we plot the average Fréchet Inception Distance (FID) (Heusel et al., 2017) for 1000 samples per timestep. The plot shows that $\mathcal{L}_{\text{top}}$ accelerates denoising and enhances the quality of $\hat{x}_0^t$ at earlier stages, working as intended. This demonstrates that $\mathcal{L}_{\text{top}}$ improves both speed and accuracy of the denoising process.

## 4.2 ABLATION STUDIES

To demonstrate the efficacy of TDN, we conduct ablation studies on the loss components and the effect of hyperparameter value changes. Appendix H includes an ablation study on the 'Encoding Network' for the topological constraint $c$. All analyses are on 0-dim Shapes dataset.

**Ablation study on $\mathcal{L}_{\text{preserve}}$ and $\mathcal{L}_{\text{denoise}}$.** In Tab. 3, we show the contribution of $\mathcal{L}_{\text{denoise}}$ and $\mathcal{L}_{\text{preserve}}$ in meeting the topological constraint. While both terms individually improve performance, $\mathcal{L}_{\text{denoise}}$ has a more significant effect. This is because $\hat{x}_0^t$, especially at larger timesteps, has several spurious structures. $\mathcal{L}_{\text{denoise}}$ works by suppressing the birth of these structures, leaving only the desired number of structures in the mask. Meanwhile, $\mathcal{L}_{\text{preserve}}$ is crucial when the model generates fewer structures than expected. Naturally, combining both terms yields the optimal performance.

**Ablation study on loss weight $\lambda$.** In Tab. 4, we show experiments with different weights for $\mathcal{L}_{\text{top}}$, including the Min-SNR ($\gamma = 5$) (Hang et al., 2023) strategy where the loss weight is a function of timestep $t$. When $\lambda = 1e-5$, TDN achieves the best performance. Nonetheless, a reasonable range of $\lambda$ always results in improvement, demonstrating the efficacy and robustness of $\mathcal{L}_{\text{top}}$.

## 5 CONCLUSION

We propose TopoDiffusionNet, the first method to integrate topology with diffusion models. Our approach generates images that preserve topology by producing masks with a specified number of structures (Betti number). Empirical results show significant improvement in preserving this topological constraint, demonstrating that our method guides the denoising process in a topology-aware manner. This paves the way for further research on topological control in image generation.

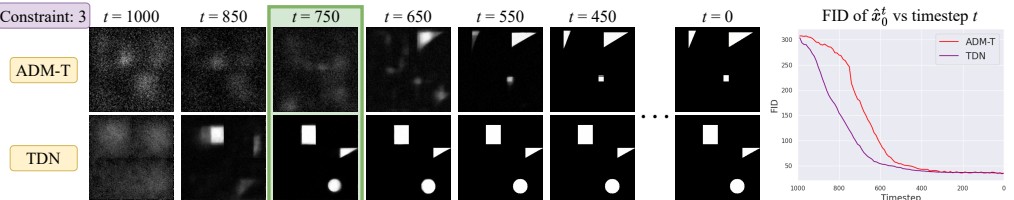

Figure 8: TDN has better FID of $\hat{x}_0^t$ at larger timesteps compared to ADM-T.

**Reproducibility Statement.** We provide experimental details regarding the datasets, baselines, evaluation metrics, and implementation in Sec. 4. Additional details of the dataset are provided in Appendix C. In Appendix D, we provide additional details about the baselines, the implementation of our method, and the computation resources used. Code available at `https://github.com/Saumya-Gupta-26/TopoDiffusionNet`

**Acknowledgments.** I would like to express my gratitude to Eisha Gupta for her valuable feedback on the writing and presentation of this paper. I also sincerely appreciate the anonymous reviewers for their time and effort in providing constructive critiques during the review and rebuttal process. This research was partially supported by the National Science Foundation (NSF) grants CCF-2144901, IIS 2212046 and SCH 2123920, the National Institute of Health (NIH) grants R01NS143143 and R01CA297843, and the Stony Brook Trustees Faculty Award.

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
