The appendix is organized as follows.

Appendix A provides the illustration of super-level sets for 1-dim topology.

Appendix B provides the illustration of $\mathcal{L}_{\text{preserve}}$ and $\mathcal{L}_{\text{denoise}}$ for 1-dim topology.

Appendix C provides additional details of the datasets.

Appendix D provides additional baseline and implementation details.

Appendix E contains 0-dim constraint-wise results on the COCO dataset.

Appendix F presents experiments on 1-dim topology where there are non-boundary (standalone) holes.

Appendix G provides qualitative results of Stable Diffusion (Rombach et al., 2022; Runway), DALL·E 3 (OpenAI, a), and AR (Phung et al., 2024) for 1-dim topological constraints.

Appendix H includes an ablation study on the 'Encoding Network' for the topological constraint $c$.

Appendix I provides a discussion on the limitations and future work of our method.

Appendix J provides further details about the persistent homology computation.

Appendix K provides FID scores for the experiments.

Appendix L presents experiments on simultaneously using 0-dim and 1-dim topological constraints as conditions.

Appendix M shows experiments using Conditional Variational AutoEncoders (cVAE), justifying the use of Diffusion Models in our work.

Appendix N provides results of using masks containing basic shapes (circles and squares) to denote the cardinality. We compare it against our TopoDiffusionNet's masks which preserve the object shape. We present results on both SD1.5 and SDXL-Turbo ControlNet backbones.

# A  PERSISTENT HOMOLOGY

From Sec. 3.2, the equivalent of Fig. 4 for 1-dimensional topology is shown in Fig. 9.

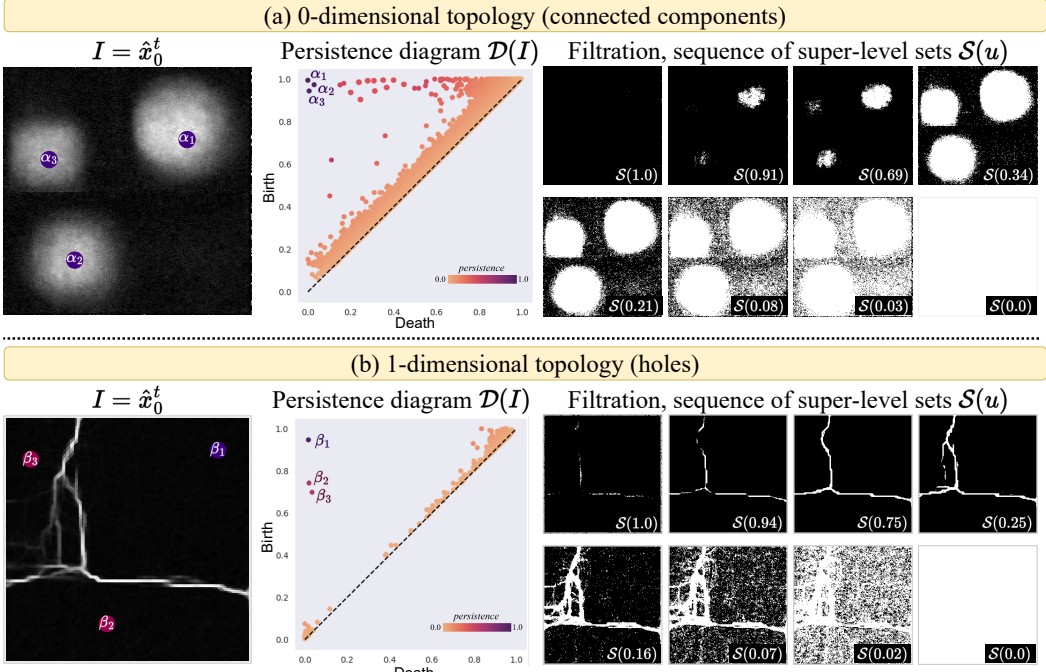

Figure 9: Illustration of persistent homology and persistence diagrams of both types of topological structures, 0-dim connected components and 1-dim holes. (a) Despite the noise, we can visually see three prominent structures $\alpha_1, \alpha_2, \alpha_3$ in $I$. In the topological space, $\alpha_1, \alpha_2, \alpha_3$ thus appear in the top-left corner of the persistence diagram $\mathcal{D}(I)$, persisting through most of the filtration $\mathcal{S}$. Similarly in (b), $\beta_1, \beta_2, \beta_3$ denote the prominent holes. All the remaining connected components and holes are noisy, persisting over a short threshold in $\mathcal{S}$, thus appearing closer to the diagonal in $\mathcal{D}(I)$. Persistence diagrams are useful to distinguish between salient and noisy structures in an image.

# B  METHODOLOGY

From Sec. 3.3, the equivalent of Fig. 5 for 1-dimensional topology is shown in Fig. 10.

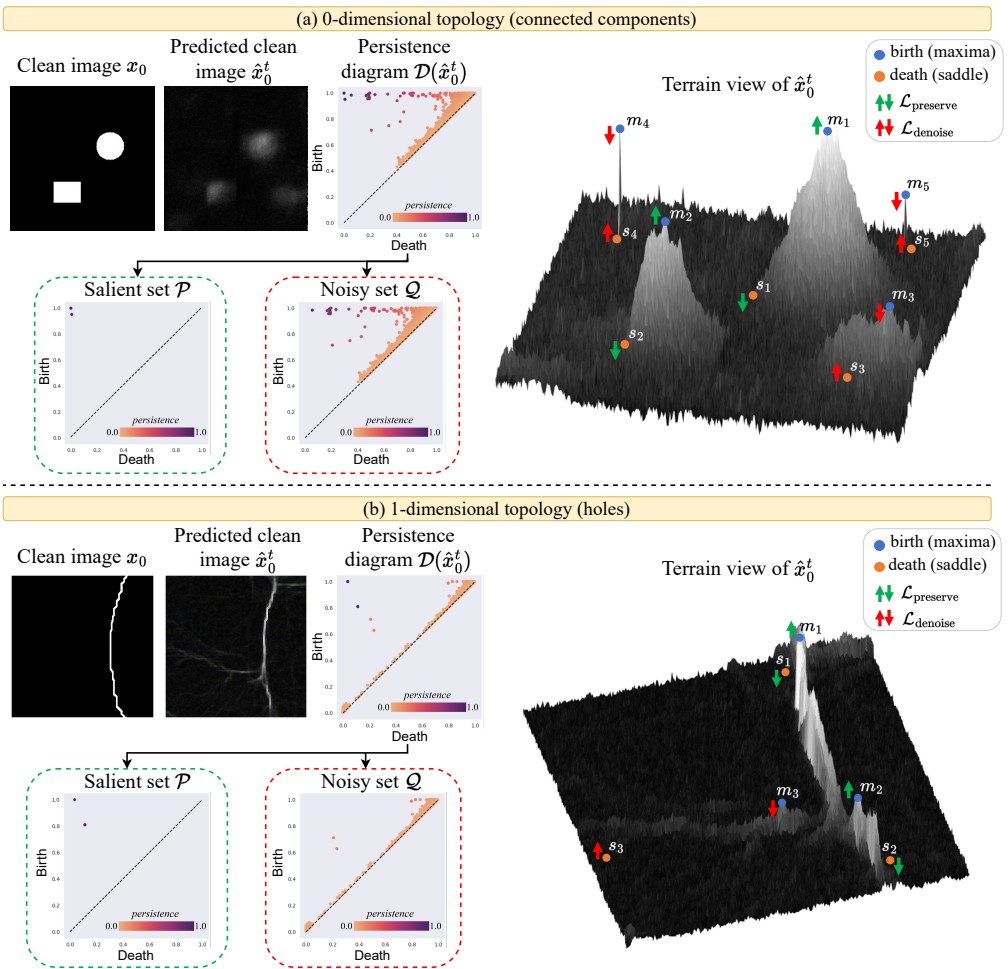

Figure 10: Illustration of $\mathcal{L}_{\text{preserve}}$ and $\mathcal{L}_{\text{denoise}}$ for 0-dim connected components and 1-dim holes, with $c = 2$ as seen in $x_0$. (a) After computing $\mathcal{D}(\hat{x}_0^t)$, we partition it into sets $\mathcal{P}$ (the top $c$ structures) and $\mathcal{Q}$ (remaining ones). For each dot $p \in \mathcal{D}(\hat{x}_0^t)$, the birth and death values respectively correspond to local maxima $m_p$ and saddles $s_p$ in $\hat{x}_0^t$. In the terrain view of $\hat{x}_0^t$, structures $(m_1, s_1)$ and $(m_2, s_2)$ belong to $\mathcal{P}$; hence optimizing $\mathcal{L}_{\text{preserve}}$ increases their saliency by increasing $\hat{x}_0^t(m_1), \hat{x}_0^t(m_2)$ and decreasing $\hat{x}_0^t(s_1), \hat{x}_0^t(s_2)$. All the remaining $n$ structures $(m_3, s_3), (m_4, s_4), \cdots, (m_n, s_n)$ belong to $\mathcal{Q}$. Optimizing $\mathcal{L}_{\text{denoise}}$ suppresses these noisy structures by decreasing $\hat{x}_0^t(m_3), \hat{x}_0^t(m_4), \cdots, \hat{x}_0^t(m_n)$ and increasing $\hat{x}_0^t(s_3), \hat{x}_0^t(s_4), \cdots, \hat{x}_0^t(s_n)$. (b) mirrors this process for holes, where $\mathcal{L}_{\text{preserve}}$ enhances the saliency of the two holes $(m_1, s_1)$ and $(m_2, s_2)$, and $\mathcal{L}_{\text{denoise}}$ suppresses the appearance of all the remaining holes like $(m_3, s_3)$.

## C    DATASET DETAILS

We provide the number of images per topological constraint $c$ used for training on each dataset in Tab. 5. For COCO (Caesar et al., 2018), since we also consider the animal class, each animal is distributed unequally across the different constraint values. For example, there were more images for 'birds' having $c = 10$ than compared to, say, 'elephant.' All the 10 animal classes are present in the dataset; they are bear, bird, cat, cow, dog, elephant, giraffe, horse, sheep, and zebra.

When curating the dataset for CREMI (Funke et al., 2016) and Google Maps (Isola et al., 2017), we manually added a (white) border to all the images. By definition of 1-dim topology, a hole is completely surrounded by a boundary. Hence, we needed to add a border to obtain the correct number of holes/regions.

Table 5: Dataset composition

| Dataset | Topological Constraint (Betti Number) | | | | | | | | | |
|---|---|---|---|---|---|---|---|---|---|---|
| | **1** | **2** | **3** | **4** | **5** | **6** | **7** | **8** | **9** | **10** |
| **Shapes** | 2K | 2K | 2K | 2K | 2K | 2K | 2K | 2K | 2K | 2K |
| **COCO (Animals)** | 520 | 517 | 503 | 297 | 176 | 85 | 64 | 38 | 30 | 27 |
| **Google Maps** | 549 | 669 | 1099 | 1220 | 1343 | 1806 | 602 | 1470 | 1054 | 662 |
| **CREMI** | 2160 | 1992 | 3726 | 3505 | 1644 | 580 | 187 | 207 | 170 | 112 |

## D    IMPLEMENTATION DETAILS

All ADM-T and TDN experiments were conducted on 1 NVIDIA RTX A6000 GPU, with a batch size of 16 and a learning rate of $2 \times 10^{-5}$. As mentioned in Sec. 3.3, our diffusion model is parameterized to predict in noise space, and we use Eq. (2) to get an estimate of the noiseless image. Although diffusion models can be parameterized to predict the noiseless state directly, we find from existing works (Hang et al., 2023; Wang et al., 2022b) that their performance is poorer compared to predicting the noise. Hence we stick to the configuration of predicting the noise. This also allows us to load pretrained weights from OpenAI (b) for our experiments instead of training from scratch.

For training ADM-T and TDN, we use the PyTorch codebase from Dhariwal & Nichol (2021)[4] and use the LSUN Bedrooms pretrained model checkpoint (OpenAI, b) to fine-tune from. To compute the birth death pairs of each topological structure, we use the Cubical Ripser (Kaji et al., 2020) library.

In Fig. 1, Fig. 6, Fig. 12, and Fig. 13, the Stable Diffusion (Rombach et al., 2022) results are generated using the Diffusers[5] library with pretrained checkpoint from Runway. For DALL·E 3, we generate images using the OpenAI API[6]. For Attention Refocusing (AR) (Phung et al., 2024), we use their publicly available codebase[7] alongwith GPT-4 (Achiam et al., 2023) from the OpenAI API to generate the layout maps. For rendering images from masks via ControlNet (Zhang et al., 2023), we use the Diffusers library with pretrained checkpoint from Lvmin Zhang (SD1.5 backbone). For Fig. 7, however, we fine-tune ControlNet on the CREMI dataset (Funke et al., 2016) so as to generate appropriate results for the corresponding text prompt.

---

[4]https://github.com/openai/guided-diffusion
[5]https://huggingface.co/docs/diffusers/en/index
[6]https://platform.openai.com/docs/api-reference/introduction
[7]https://github.com/Attention-Refocusing/attention-refocusing

## E CONSTRAINT-WISE RESULTS

In Fig. 11, we plot the accuracy of the different methods on the COCO dataset. For smaller object counts, that is, $c <= 5$, the performance of each method is better than $c > 5$. At higher object counts, although the accuracy of each method reduces, TDN still significantly outperforms the baselines.

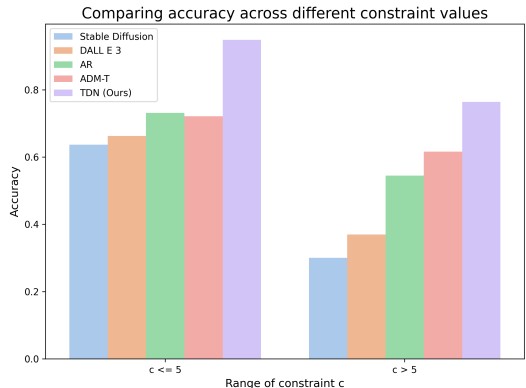

Figure 11: Accuracy results on low and high object counts (COCO dataset).

## F ADDITIONAL EXPERIMENTS ON HOLES

In the main paper, we show experiments on 1-dim topology using the CREMI (Funke et al., 2016), and Google Maps (Isola et al., 2017) datasets. In these datasets, it makes sense that the holes are with respect to the image frame/boundary and span the whole image. However, TDN can handle 1-dim holes in general, not just those with respect to the boundary. To demonstrate this, we conduct experiments on standalone holes (not relative to the boundary), and present the results in Tab. 6 and Fig. 12. We generate a synthetic dataset of circular rings (similar to the Shapes dataset) to train TDN, and use the prompt 'donuts' in ControlNet to render the image. While generating $c$ donuts could also be achieved using the 0-dim topological constraint, this experiment highlights the 1-dim generalizability of TDN.

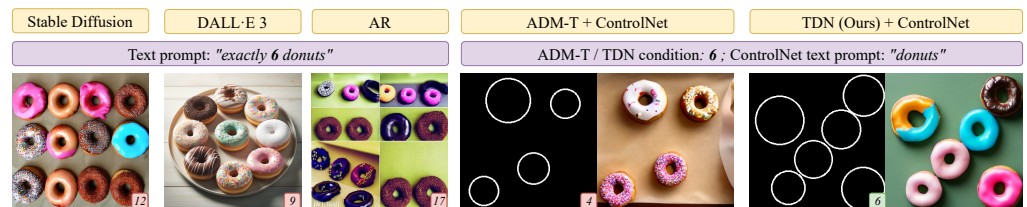

Figure 12: Results on standalone holes (holes not relative to the image frame/boundary). Number of donuts within each mask/image is noted in its bottom-right inset.

Table 6: Standalone holes

| Method | Accuracy ↑ | F1 ↑ |
|---|---|---|
| ADM-T | $0.79 \pm 0.12$ | $0.81 \pm 0.11$ |
| TDN (Ours) | $\mathbf{0.95 \pm 0.03}$ | $\mathbf{0.96 \pm 0.02}$ |

## G ADDITIONAL 1-DIM QUALITATIVE RESULTS

In Fig. 13, we show qualitative results of pretrained Stable Diffusion (Rombach et al., 2022), DALL·E 3 (OpenAI, a) and Attention Refocusing (AR) (Phung et al., 2024) for the same 1-dim constraints shown in Fig. 7 of the main paper. As 1-dim topology is hard to describe in words, we tried a lot of variations for the text prompts, and show the results from the best ones in the figure.

As the pretrained Stable Diffusion, DALL·E 3, and AR are not trained on Electron Microscopy images of cell neurons, their inaccurate results are understandable. For generating roads, however, these methods do generate visually appropriate images but struggle to maintain the correct number of holes/regions. AR additionally tends to generate images that appear to be divided into separate, unconnected sections. This is due to the use of layout maps in the reverse process. All these methods are limited to generating images with 0-dim topology (i.e., distinct objects), and do not extend to 1-dim topology. This shortcoming motivates our work on TopoDiffusionNet.

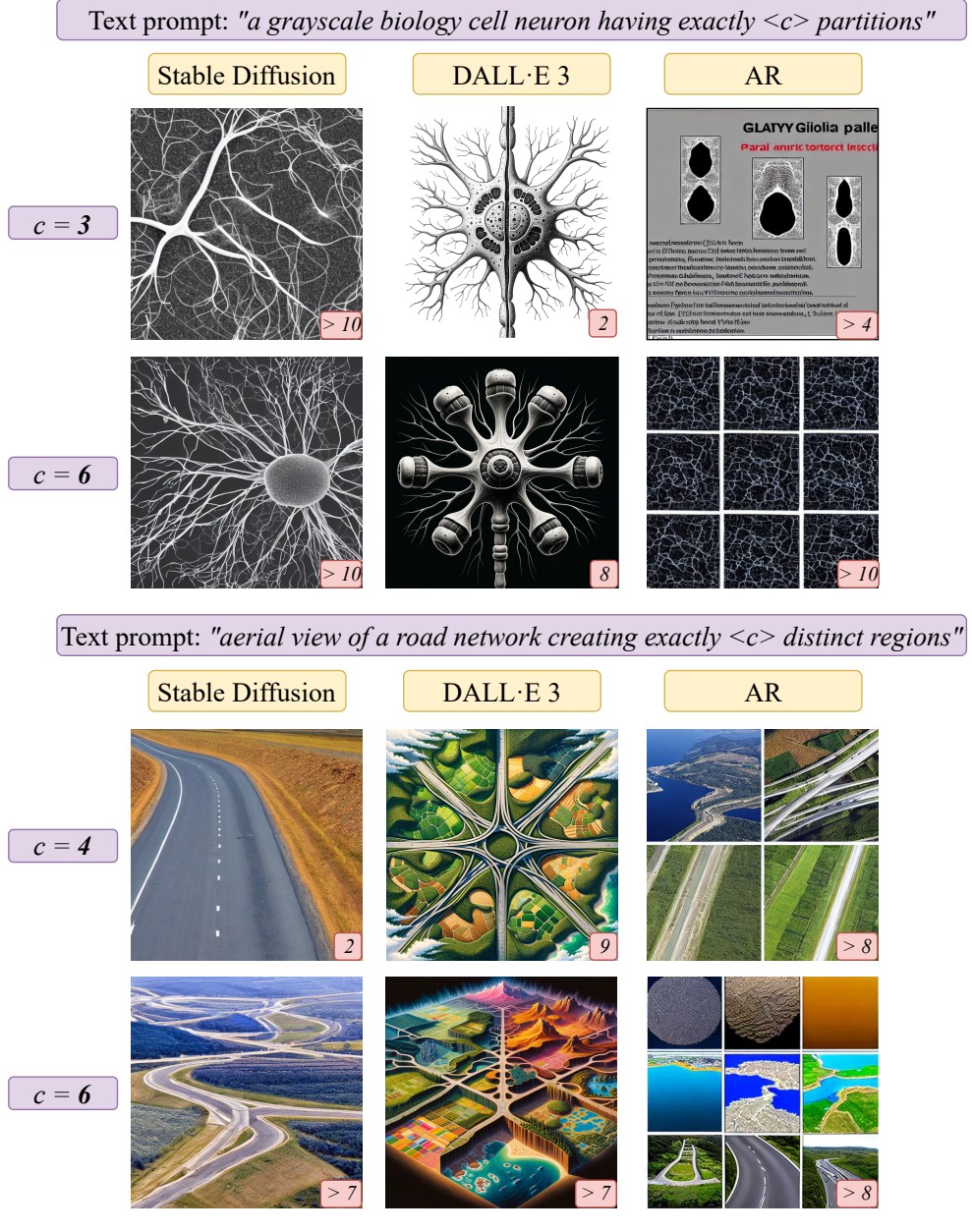

Figure 13: Qualitative results for 1-dim topological constraint. Stable Diffusion, DALL·E 3 and AR take text prompts as input (purple box). Rows 1-2: Results equivalent to CREMI. Rows 3-4: Results equivalent to Google Maps. Number of holes within each image is noted in its bottom-right inset.

# H    ADDITIONAL ABLATION STUDY

We conduct an additional ablation study apart from the ones presented in the main paper.

**Ablation study of the encoding network.** In TDN, we use the topological constraint $c$ as a condition. We do this by first obtaining an embedding of $c$ via an Encoding Network (Fig. 3), and then passing it to all the residual blocks in the denoising model.

In the results reported in the main paper, the Encoding Network is composed of a few linear layers. We use 'LL' to denote this configuration.

Another way to configure the Encoding Network is to use the Transformer sinusoidal position embedding (Vaswani et al., 2017). Let 'PE' denote this configuration. It uses the same code as the network generating the sinusoidal timestep embedding.

We show results comparing LL and PE in Tab. 7 when using our proposed objective function $\mathcal{L}_{\text{top}}$. We find that both configurations have comparable performance, with LL slightly outperforming PE.

Table 7: Ablation study on Encoding Network for TDN

| Dataset | Encoding | Accuracy ↑ | Precision ↑ | F1 ↑ |
|---------|----------|------------|-------------|------|
| **Shapes** | LL | **0.9478 ± 0.0420** | **0.9499 ± 0.0492** | **0.9488 ± 0.0370** |
|  | PE | 0.9011 ± 0.0730 | 0.9132 ± 0.0683 | 0.9033 ± 0.0385 |
| **COCO** | LL | **0.8557 ± 0.0805** | **0.8670 ± 0.0636** | **0.8613 ± 0.0970** |
| **(Animals)** | PE | 0.8395 ± 0.0952 | 0.8436 ± 0.1152 | 0.8411 ± 0.1014 |

# I    DISCUSSION

**Limitations.** Presently, TDN requires at least a few samples of each constraint in the training data and is not guaranteed to extrapolate to unseen constraints. This limitation stems from the broader challenges associated with state-of-the-art diffusion models, which require a large amount of training data. Exploring the numeric relationships between the different constraints, instead of treating them independently, has the potential to generalize to unseen constraints.

**Future work.** Our proposed TDN currently uses persistent homology to control the number of objects (in 0-dim) and the number of holes (in 1-dim). However, persistent homology can theoretically be extended to higher dimensions, as persistence diagrams can capture topological features in arbitrary dimensions. For future work, we are looking into graph network generations, as well as 3D applications, where we can control not just connected components (0-dim) and holes (1-dim), but also voids (2-dim). 3D point clouds and volumetric medical imaging data are important applications where maintaining topology is crucial in generating realistic synthetic data.

## J  DETAILS ON PERSISTENT HOMOLOGY COMPUTATION

We expand on Sec. 3.2 regarding the persistent homology and persistence diagram (PD) computation. As mentioned in the implementation details (Appendix D), we use the Cubical Ripser library Kaji et al. (2020) for the homology computation. Their manuscript has details about the optimized algorithms they implement for the computation. Here, we provide a summarized version of the details.

Consider a 2D image $I \in \mathbb{R}^2$. We consider the 0-dim case in which we track the birth and death of connected components (CC). As we construct a super-level filtration $\mathcal{S}$, we start thresholding an image $I$ starting from the maximum value to the minimum value. For computational purposes, we do not need to consider all threshold values $u \in \mathbb{R}$, rather, since the image has size $H \times W$, the maximum number of unique values in the image is $H * W$. Hence we need to consider atmost $H * W$ values of threshold $u$ to determine the birth and death times of all the CC in the image.

First, preprocessing is done to store all the intensity values in $I$ along with their $(x, y)$ location in decreasing order in a sorted data structure. This allows for faster identification of the next threshold value as well as the pixels that get newly included in the super-level set. Second, we utilize the standard Union-Find data structure Tarjan (1975), which maintains a collection of disjoint sets. This data structure supports the operation FIND((x,y)), which finds the highest-value representative of the CC containing the pixel $(x, y)$. It also supports the operation UNITE((x,y), (v,w)), which unites the CCs represented by root pixels $(x, y)$ and $(v, w)$ and, assuming $I(x, y) > I(v, w)$, making $(x, y)$ the representative of the merged components.

At each threshold value $u$, we use the sorted data structure to identify which pixels are included in the set $\mathcal{S}(u)$. FIND is called on all these newly added pixels to determine if they belong to any existing CC; if not, they are considered roots of a new CC, with a new entry to the list of disjoint sets in the union-find data structure. Furthermore, this results in the creation of a dot $(u, -\infty)$ in the persistence diagram whose birth time is $u$, and a default death time of $-\infty$. Then, UNITE is called for every pair of roots in the disjoint set list, to determine if any of the sets are in fact merged. Consider if sets with roots $(x, y)$ and $(v, w)$ are merged, with $I(x, y) > I(v, w)$. In that case, CC with root $(v, w)$ has now 'died', and so the persistence dot whose birth time was $I(v, w)$ in the diagram, will now have an updated death time of $u$. Hence, $(I(v, w), u)$ will replace the old dot $(I(v, w), -\infty)$ in the persistence diagram.

We show a walkthrough of this in Fig. 14. We start with an empty union-find data structure. In (a), we start with $u = 5$, as 5 is the maximum value in the image. We call FIND and update the union-find data structure with a new entry. Additionally, the persistence diagram (PD) in (e) has an entry for $(5, -\infty)$. This CC is denoted as $\alpha_1$. Next, in (b) $u = 4$, all the blue and red pixels are newly added to this filtration. FIND is called on each of them. The blue pixels indicate that they were found to be a part of an existing CC. The red pixels indicate that no parent CC was found, and hence are considered as creating new CCs. The PD includes two new dots $\alpha_2(4, -\infty)$ and $\alpha_3(4, -\infty)$. Next, as intensity value 3 is not present in $I$, we do not need to compute $\mathcal{S}(3)$. The sorted data structure directly leads us to the next $u$ value which is $u = 2$ in (c). FIND is called on all the blue pixels. UNITE is called on all pairs of root CC nodes, resulting True for the fact that components $\alpha_1$ and $\alpha_2$ have now merged. Since the birth time of $\alpha_1$ is larger, the component $\alpha_2$ 'dies' and becomes a part of $\alpha_1$. Thus, the persistent dot is updated from $\alpha_2(4, -\infty)$ to $\alpha_2(4, 2)$ as it was born at $u = 4$ has died at $u = 2$. Similarly in (d), at $u = 1$, FIND is called on all the blue pixels. Then UNITE ($\alpha_1$, $\alpha_3$) is called, showing that they are actually merged. Thus, the persistent dot is updated from $\alpha_3(4, -\infty)$ to $\alpha_3(4, 1)$ as $\alpha_3$ was born at $u = 4$ and has now died at $u = 1$. Thus, the final persistent dots we end up with are $\alpha_1(5, -\infty)$, $\alpha_2(4, 2)$, $\alpha_3(4, 1)$ as shown in (e).

For 1-dimensional and higher features (e.g., loops and voids), while we do not go into details here, the Cubical Ripser library constructs a sparse coboundary matrix that encodes relationships between pixels. This matrix is reduced to upper-triangular form using column operations, optimizing the identification of the birth and death of cycles. Each non-zero pivot in the reduced matrix corresponds to a topological feature. This computational framework ensures the efficient generation of persistence diagrams, leveraging union-find for 0-dim and matrix operations for higher dimensions. More details are available in the Cubical Ripser Kaji et al. (2020) paper.

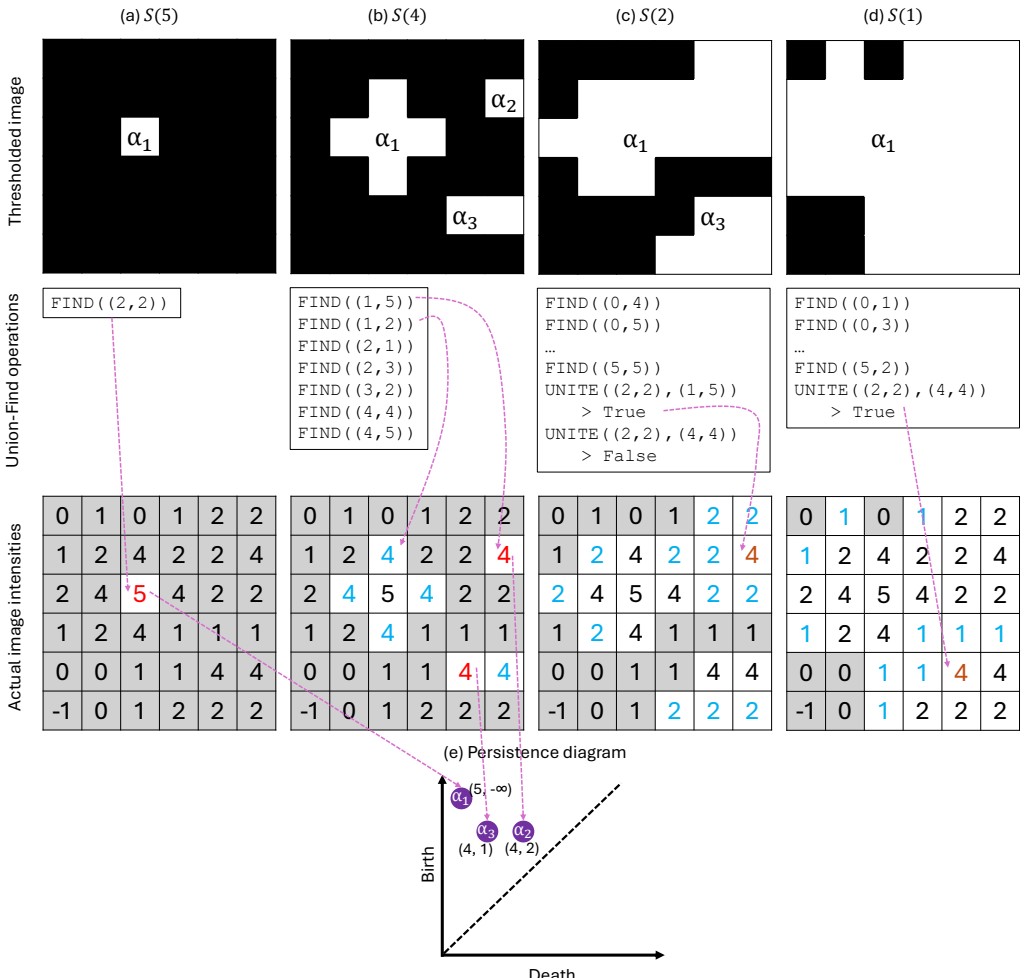

Figure 14: Figure inspired from Kaji et al. (2020). The figure shows a sample 2D image, thresholded at different values. (a)-(d) shows the sequence of super-level sets $\mathcal{S}(u)$, which is nothing but thresholding the image at the value $u$. It shows how different connected components (CC) are created or destroyed as $u$ changes. White-background cells indicate pixels included in the set $\mathcal{S}(u)$. Red indicates root pixels causing the birth of a CC. Blue indicates pixels that are newly added to the set $\mathcal{S}(u)$ but are absorbed in existing CCs. Brown indicates root pixels whose CC has died, having merged with another CC whose root had a higher birth time. (e) shows the corresponding 0-dim persistence diagram.

# K    QUALITY OF GENERATED MASKS

We provide FID scores across all the datasets in Tab. 8. Since our method TDN focuses on generating masks, we report the FID of the mask [FID (Mask)] and the FID of the images [FID (Image)] generated by ControlNet Zhang et al. (2023) when using these masks as condition.

Our analysis shows that the quality of the masks generated by TDN closely resembles the true masks, that is, the ground truth (GT) masks annotated by humans, as evidenced by the consistently low FID (Mask) scores across all datasets. Additionally, to evaluate the impact on final image quality, we compare the FID (Image) metric between two scenarios: ControlNet using GT masks (ControlNet + GT) versus ControlNet using TDN-generated masks. The results demonstrate comparable FID scores, indicating that using TDN-generated masks as conditioning does not degrade the image quality. In other words, TDN-generated masks are as good as GT masks for generating real images.

Results from Tab. 1 and Tab. 8 highlight that TDN achieves improved topological control while maintaining the overall visual quality of the generated images.

Table 8: **Comparison of FID (Mask) and FID (Image) against baselines across all datasets.** For Shapes and Google Maps datasets, we cannot report FID (Image) as there is no ground truth image dataset to compare to. For COCO and CREMI, we report FID (Image) against the respective dataset images. ControlNet + GT indicates that ControlNet is using the GT masks as the condition. ControlNet + FT indicates that it has been fine-tuned on the CREMI dataset to generate similarly textured images. **Bold** denotes the best results, while *italics* denotes the second best. Note that ControlNet + GT has been included as the upper bound performance; realistically, it cannot be used during inference as the GT masks are not available

| Dataset | Method | FID (Mask) $\downarrow$ | FID (Image) $\downarrow$ |
|---|---|---|---|
| **Shapes** | ADM-T | 0.092 | - |
| | TDN (Ours) | **0.068** | - |
| **COCO (Animals)** | Stable Diffusion | - | 29.41 |
| | DALL·E 3 | - | **17.49** |
| | AR | - | 35.05 |
| | ADM-T | 0.267 | 21.72 |
| | TDN (Ours) | **0.222** | *21.28* |
| | ControlNet + GT | - | 20.94 |
| **Google Maps** | ADM-T | 0.198 | - |
| | TDN (Ours) | **0.156** | - |
| **CREMI** | Stable Diffusion | - | 48.18 |
| | DALL·E 3 | - | 54.72 |
| | AR | - | 69.86 |
| | ADM-T | 0.518 | 3.322 |
| | TDN (Ours) | **0.467** | **3.286** |
| | ControlNet + GT + FT | - | 3.126 |

## L  USING 0-DIM AND 1-DIM SIMULTANEOUS TOPOLOGICAL CONSTRAINTS

In our work, we focused on using either 0-dim (number of objects) or 1-dim (number of holes) constraints individually, depending on the property of the dataset. In this section, we conduct experiments on synthetic data using both 0-dim and 1-dim constraints simultaneously. The constraint is of the form $(a, b)$ where $a$ denotes 0-dim while $b$ denotes 1-dim. Both are first separately encoded and then added together to be fed as a condition. Sample masks generated by TDN along with ControlNet-rendered images are shown in Fig. 15. We report quantitative results in Tab. 9.

We find that TDN is capable of handling these joint constraints. While the performance is slightly lower compared to using each constraint individually, TDN still significantly outperforms ADM-T across all metrics. This shows that persistent homology (PH) can enhance the performance of the base diffusion model, even under multi-constraint scenarios. This finding opens up interesting possibilities for future work in handling even richer combinations of topological constraints.

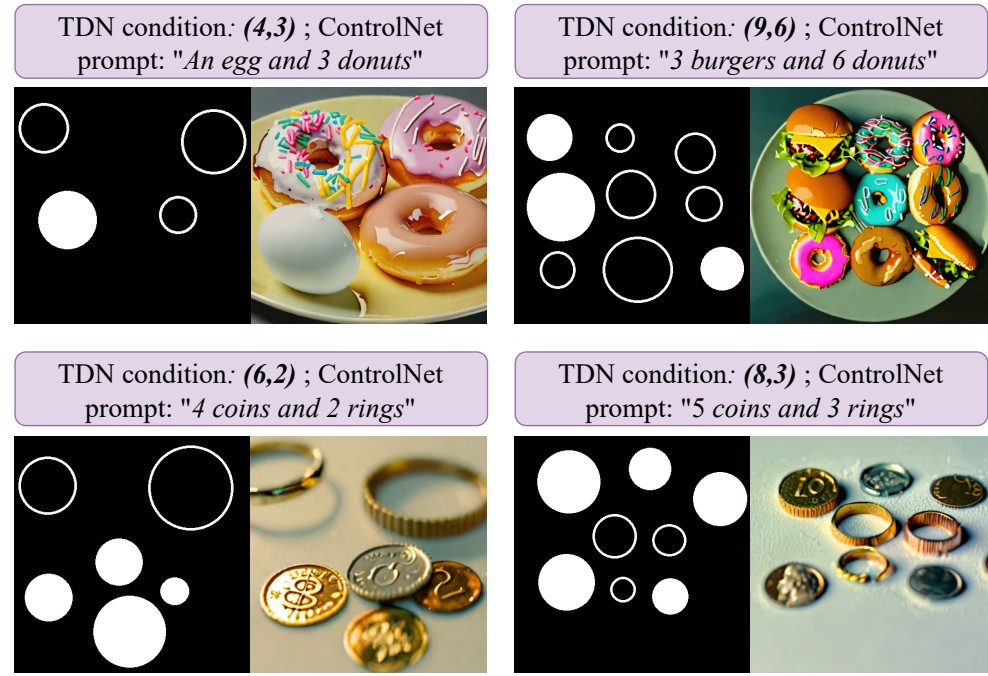

Figure 15: Masks generated by TDN using both 0-dim and 1-dim topological constraints simultaneously. Condition $(a, b)$ denotes $a$ for 0-dim (#objects) and $b$ for 1-dim (#holes). Images are generated via ControlNet by using mask as a condition.

Table 9: Quantitative performance of using both 0-dim and 1-dim topological constraints simultaneously. The FID (Mask) is reported once for each method. Best results are highlighted in **bold**

| TopoDim | Method | Accuracy ↑ | Precision ↑ | F1 ↑ | FID (Mask) ↓ |
|---|---|---|---|---|---|
| 0-dim | ADM-T | $0.7383 \pm 0.1305$ | $0.7997 \pm 0.1268$ | $0.7677 \pm 0.1229$ | 0.1279 |
|  | TDN (Ours) | $\mathbf{0.9183 \pm 0.0731}$ | $\mathbf{0.9338 \pm 0.0993}$ | $\mathbf{0.9261 \pm 0.0906}$ | **0.0982** |
| 1-dim | ADM-T | $0.7616 \pm 0.0905$ | $0.7892 \pm 0.1129$ | $0.7752 \pm 0.1082$ | - |
|  | TDN (Ours) | $\mathbf{0.9233 \pm 0.0705}$ | $\mathbf{0.9492 \pm 0.0913}$ | $\mathbf{0.9360 \pm 0.0720}$ | - |

## L.1 GENERATING SPECIFIC TOPOLOGY LIKE DOUBLE ANNULUS

As mentioned above, we represent the joint constraints as pairs $(a, b)$, where $a$ specifies the number of connected components (0-dim) and $b$ specifies the number of holes (1-dim). A double annulus consists of concentric circles, with $(a, b) = (2, 2)$. Our loss, based on homology, cannot distinguish whether two loops are positioned concentric or not, however, it can generate structures that are homotopy equivalent to the double annulus. As shown in Fig. 16, by sampling for the $(2, 2)$ condition, TDN can generate nested circles (visually the closest constraint to a double annulus) and other homotopy equivalent structures. This experiment demonstrates that our method can handle complex topological specifications through the combination of constraints, though geometric relationships (like equidistance and concentricity) remain an interesting direction for future work.

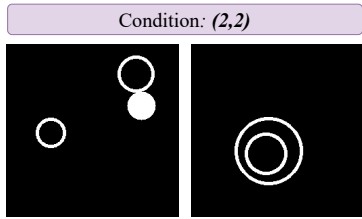

Figure 16: Sample images generated by TDN when using both 0-dim and 1-dim topological constraints simultaneously. With constraints $(a, b) = (2, 2)$, we can generate nested circles which are the closest to achieving a double annulus (two concentric circles) using our method.

## M    COMPARISON AGAINST CVAE

In this section, we justify the motivation behind using diffusion models instead of other generative models like Conditional Variational Autoencoders (CVAE) Sohn et al. (2015). We conduct experiments using CVAE as the generative model for the same task of generating masks. Although we trained on multiple image resolutions, CVAE performed poorly on larger image sizes. Hence we report results on $64 \times 64$ images, using a latent embedding dimension of $128$. We show results in Tab. 10 and Fig. 17 for both standard CVAE and a version incorporating our proposed loss $\mathcal{L}_{\text{top}}$.

Our experimental results show significant limitations of the CVAE approach. From Fig. 17, we see that the quality of the masks generated by CVAE, with and without $\mathcal{L}_{\text{top}}$, is severely degraded. From Tab. 10, it has a significantly higher FID compared to our TDN masks, indicating a lower resemblance to the true masks. Additionally, CVAE also struggles to preserve topological constraints, as evidenced by the low performance on evaluation metrics like Accuracy and F1. While CVAE $+\mathcal{L}_{\text{top}}$ slightly improves performance, it remains far weaker than diffusion models. Specifically, in the 0-dim, CVAE generates fragmented objects and fails to preserve their overall shapes. In 1-dim, it fails to generate connected structures. Although mask images seem easy because they are binary images, they are in fact a challenge to generate. This is because of the difference in the number of objects, their varied shapes and sizes, and unconstrained spatial locations (eg: not fixed to the center of the image). Similarly, the diverse orientation and connection patterns make the 1-dim datasets challenging. Such complexities make it difficult for the CVAE to generalize to these datasets.

Using diffusion models is critical to the success of TDN. First, the base diffusion model ADM-T effectively captures spatial arrangements and object shapes (in 0-dim), and connectivity patterns in 1-dim. However, it struggles with the number of objects/holes. This is where our persistent homology loss brings big improvements. Preserving the number of structures requires heavy global reasoning and detail. In contrast, CVAE's bottleneck layer compresses information, losing significant information in the image space. This limits their ability to preserve object shape and enforce strict topological constraints. The significant gain of our loss in diffusion models is owing to the gradual denoising steps, as well as intact image resolution. These experiments highlight the necessity of diffusion models, making them critical to the success of our method in topological control.

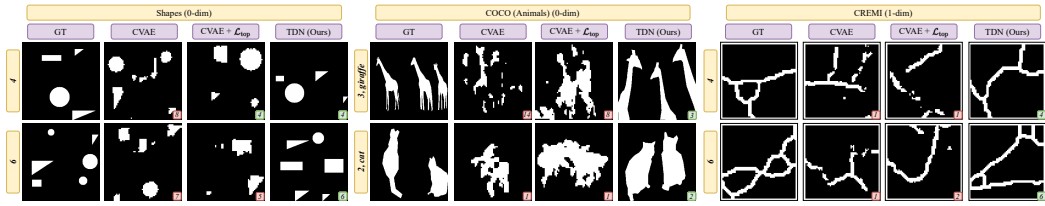

Figure 17: Qualitative comparison of using CVAE and diffusion models as the generative model. Ground truth (GT) denotes the true masks used to train the models

Table 10: Quantitative comparison of using CVAE and diffusion models as the generative model. We include ADM-T and TDN results from Tab. 1 for convenience. Best results are in **bold**

| Dataset (TopoDim) | Method | Accuracy ↑ | Precision ↑ | F1 ↑ | FID (Mask) ↓ |
|---|---|---|---|---|---|
| **Shapes (0-dim)** | CVAE Sohn et al. (2015) | $0.3133 \pm 0.1321$ | $0.3235 \pm 0.1984$ | $0.3078 \pm 0.1443$ | 2.231 |
| | CVAE $+\mathcal{L}_{\text{top}}$ | $0.3816 \pm 0.0919$ | $0.3444 \pm 0.1276$ | $0.3220 \pm 0.1365$ | 1.981 |
| | ADM-T | $0.7500 \pm 0.1889$ | $0.7809 \pm 0.1582$ | $0.7651 \pm 0.1210$ | 0.092 |
| | TDN (Ours) | $\mathbf{0.9478 \pm 0.0420}$ | $\mathbf{0.9499 \pm 0.0492}$ | $\mathbf{0.9488 \pm 0.0370}$ | **0.068** |
| **COCO (0-dim)** | CVAE Sohn et al. (2015) | $0.2442 \pm 0.0410$ | $0.2696 \pm 0.0285$ | $0.2562 \pm 0.0318$ | 4.208 |
| | CVAE $+\mathcal{L}_{\text{top}}$ | $0.3094 \pm 0.0967$ | $0.3342 \pm 0.0729$ | $0.3213 \pm 0.1165$ | 4.083 |
| | ADM-T | $0.6685 \pm 0.1485$ | $0.6917 \pm 0.1079$ | $0.6799 \pm 0.1931$ | 0.267 |
| | TDN (Ours) | $\mathbf{0.8557 \pm 0.0805}$ | $\mathbf{0.8670 \pm 0.0636}$ | $\mathbf{0.8613 \pm 0.0970}$ | **0.222** |
| **CREMI (1-dim)** | CVAE Sohn et al. (2015) | $0.2785 \pm 0.4296$ | $0.2267 \pm 0.1156$ | $0.2499 \pm 0.1391$ | 5.971 |
| | CVAE $+\mathcal{L}_{\text{top}}$ | $0.3267 \pm 0.2834$ | $0.3091 \pm 0.1079$ | $0.3176 \pm 0.1447$ | 5.751 |
| | ADM-T | $0.5357 \pm 0.1879$ | $0.4777 \pm 0.1797$ | $0.4881 \pm 0.1571$ | 0.518 |
| | TDN (Ours) | $\mathbf{0.7785 \pm 0.1901}$ | $\mathbf{0.8142 \pm 0.1925}$ | $\mathbf{0.7959 \pm 0.1659}$ | **0.467** |

## N    COMPARISON AGAINST BASIC MASKS

To justify the importance of our approach, we conduct experiments using simple shapes (circles and squares) as conditioning masks for ControlNet Zhang et al. (2023). Quantitative results are shown in Tab. 11, and qualitative examples are provided in Fig. 18.

Our experiments reveal several limitations when using simplified shapes as conditioning masks, as they fail to provide sufficient spatial and structural guidance. This setup is comparable to AR Phung et al. (2024) (shown in Tab. 1 and Fig. 6 of the main paper), which uses rectangular bounding boxes with an additional attention step. From our observations, we detect multiple failure modes. First, from Fig. 18, we see that such conditions tend to introduce visual artifacts in the generated images. The images are often fragmented, with objects isolated within assigned areas, leading to a divided and visually disjointed image. Second, the basic shapes also fail to constrain the number of objects within them, often resulting in 0 objects or multiple objects per shape (see the 'cats' row). Finally, for large animals like zebras, ControlNet struggles to complete the image beyond the boundary of the shape—while the zebra texture is present within the shape, the overall image is not as desired. Similarly, for even larger animals like giraffes, the discrepancy between the basic shape and the natural object proportions leads to incomplete or distorted generations.

In contrast, the masks generated by TDN provide tighter spatial and structural control by offering sufficient detail to guide the network effectively. This results in correct object counts and better visual quality, as evidenced by superior FID scores and count metrics. TDN's masks strike a balance between oversimplification and unnecessary complexity, as generative models are powerful enough to fill in finer textural details.

Our results highlight that relying solely on a text prompt and simplified masks is insufficient for the model to generate visually coherent images. Instead, meaningful masks like those generated by TDN are essential for ensuring reliable control over cardinality and topology in generative models.

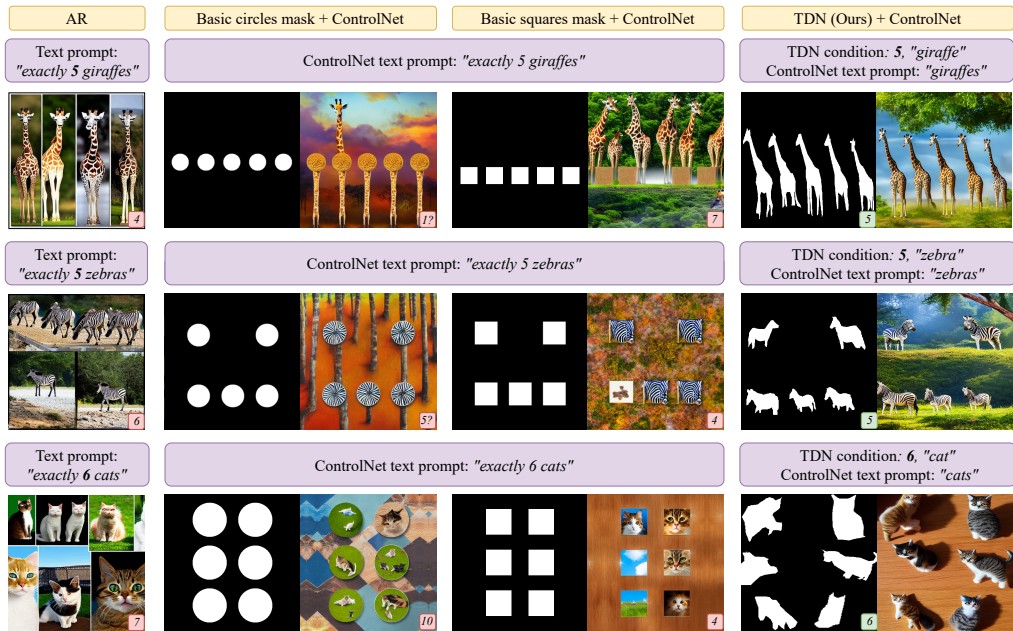

Figure 18: Qualitative comparison of using different conditioning types, particularly masks containing basic shapes like circles/squares to guide cardinality. ControlNet uses the SD1.5 backbone (Lvmin Zhang).

Table 11: Quantitative comparison of using different conditioning types for the COCO dataset. We include AR (Phung et al., 2024) and TDN results from Tab. 1 for convenience. Best results in **bold**

| Method / Mask condition type | Accuracy ↑ | Precision ↑ | F1 ↑ | FID (Image) ↓ |
|---|---|---|---|---|
| Basic circles mask + ControlNet | 0.2972 ± 0.3592 | 0.3194 ± 0.3046 | 0.3079 ± 0.2208 | 49.47 |
| Basic squares mask + ControlNet | 0.3123 ± 0.2114 | 0.3459 ± 0.2209 | 0.3282 ± 0.2618 | 47.16 |
| AR (Bounding box w/ attention) | 0.6379 ± 0.2062 | 0.7360 ± 0.1658 | 0.6611 ± 0.1851 | 35.05 |
| TDN (Ours) | **0.8557 ± 0.0805** | **0.8670 ± 0.0636** | **0.8613 ± 0.0970** | **21.28** |

### N.1 USING SDXL-TURBO BACKBONE FOR CONTROLNET

The ControlNet results generated in this paper have used the SD1.5 backbone (Lvmin Zhang). In this sub-section, we generate ControlNet results using the newer SDXL-Turbo backbone[8] (Sauer et al., 2025). The results are in Fig. 19, Fig. 20, and Fig. 21.

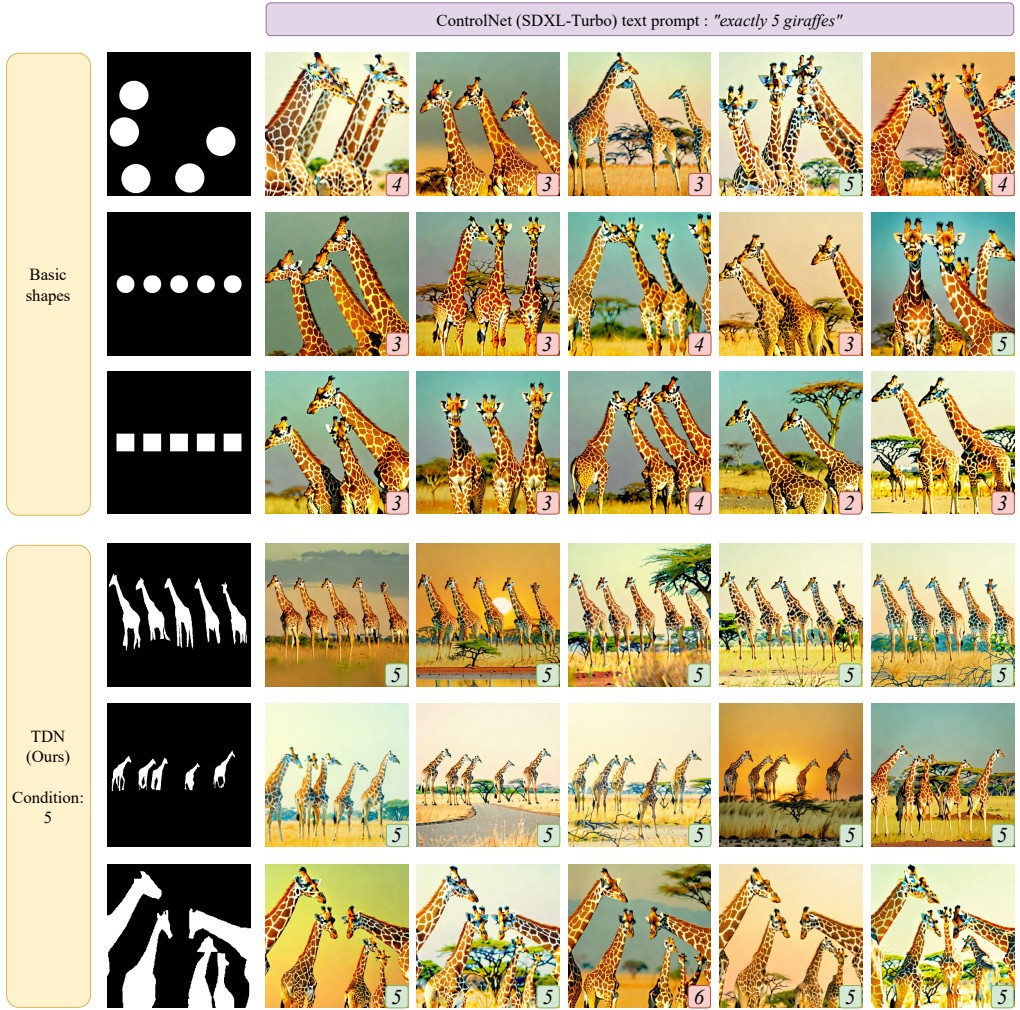

Figure 19: Generating exactly 5 giraffes. Qualitative comparison of using different conditioning types, such as masks containing basic shapes like circles/squares to guide cardinality. ControlNet uses the SDXL-Turbo backbone (Sauer et al., 2025).

---

[8] https://huggingface.co/stabilityai/sdxl-turbo

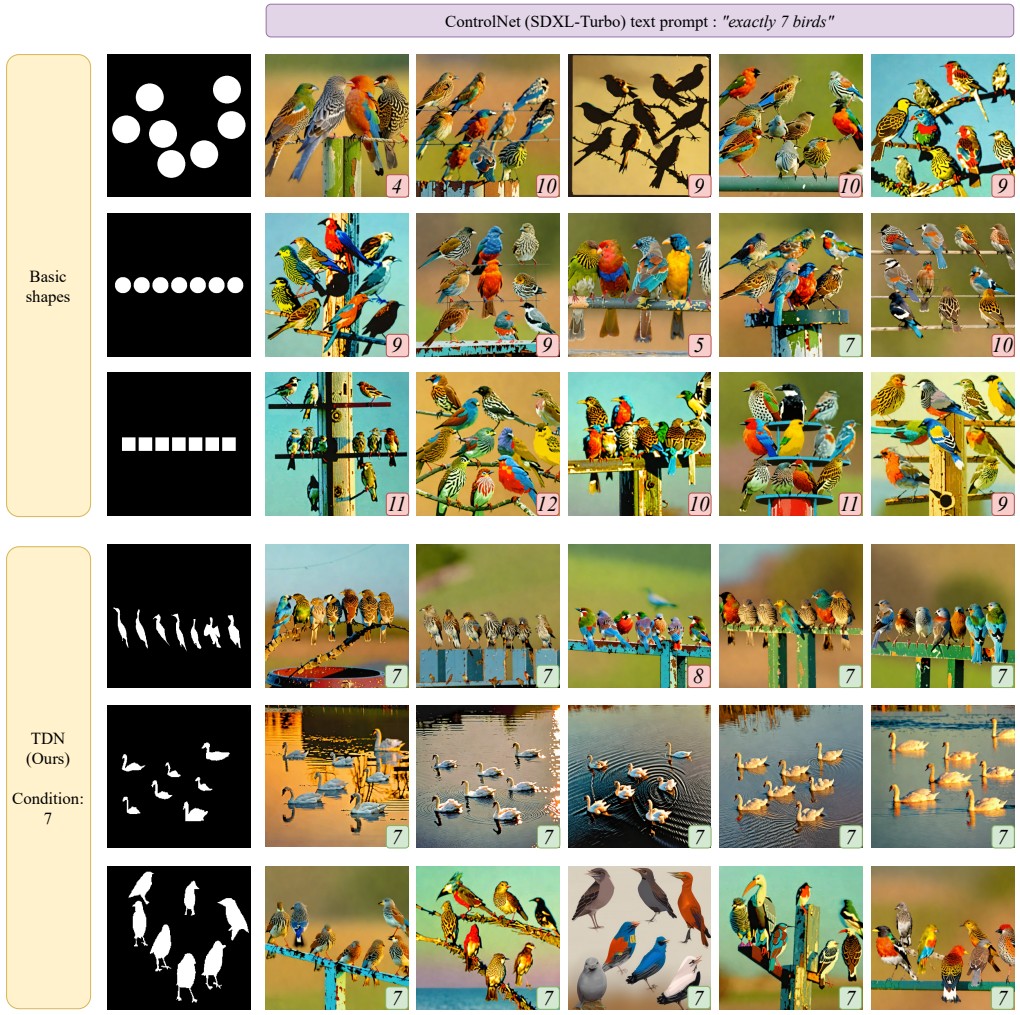

Figure 20: Generating exactly 7 birds. Qualitative comparison of using different conditioning types, such as masks containing basic shapes like circles/squares to guide cardinality. ControlNet uses the SDXL-Turbo backbone (Sauer et al., 2025).

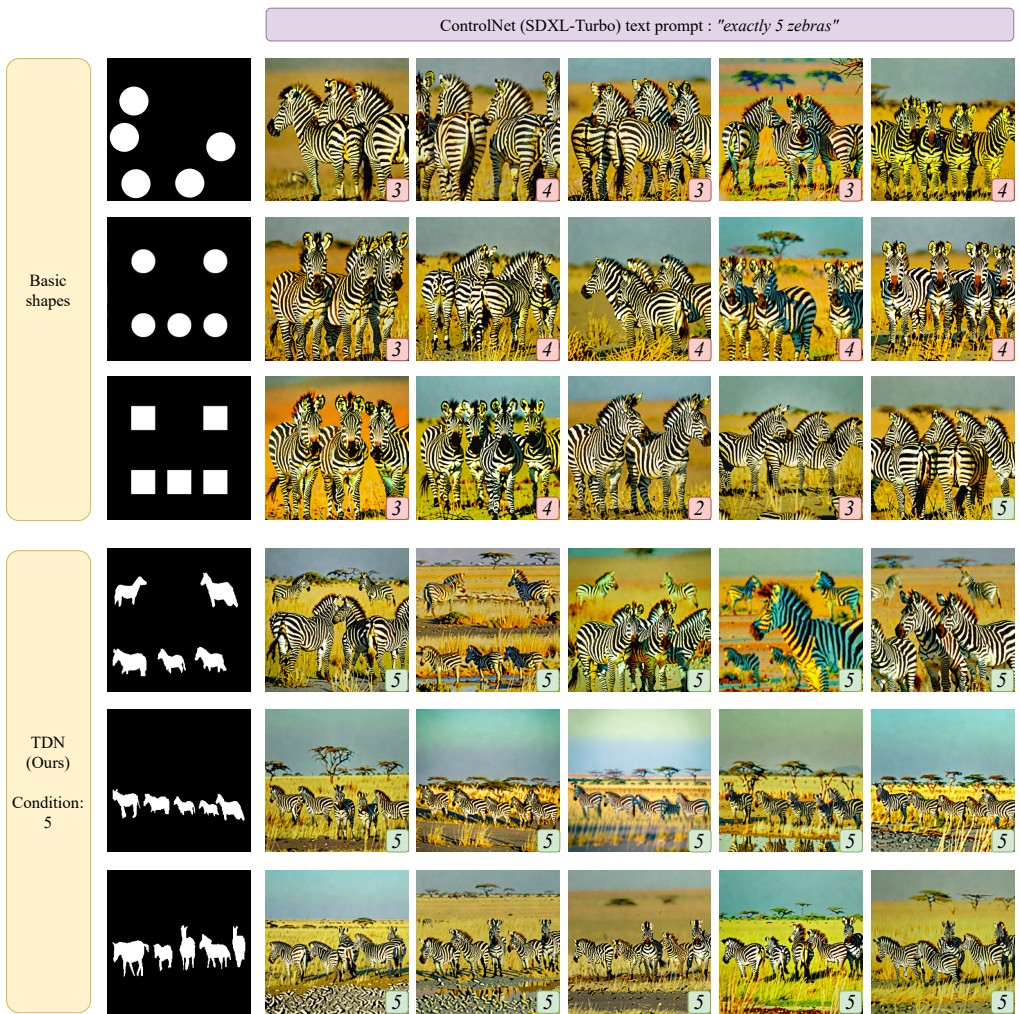

Figure 21: Generating exactly 5 zebras. Qualitative comparison of using different conditioning types, such as masks containing basic shapes like circles/squares to guide cardinality. ControlNet uses the SDXL-Turbo backbone (Sauer et al., 2025).