# OpenReview forum: "TopoDiffusionNet: A Topology-aware Diffusion Model"
_ICLR.cc/2025/Conference — ICLR 2025 Poster_

### Official Review · Reviewer_mfGQ · 2024-10-24

**Soundness:** 3
**Presentation:** 3
**Contribution:** 3
**Rating:** 8
**Confidence:** 3

**Summary:**

This paper proposes a method to control the topology of images generated by the diffusion model. By combining the persistent homology theory, a topological mask is generated with noise, and it is fed to the diffusion model as a condition together with the time steps, achieving more accurate topological control of image generation.

**Strengths:**

The authors claim that they are the first to combine topological and diffusion models on image generation. Although I cannot be sure of this, I think it is indeed a problem that needs to be solved urgently, and this paper provides a possible solution to it.

The combination of persistent homology and diffusion modeling is new to me, and although the combination way is simple, it appears to be effective.

This paper is well written, with rich comparative data and ablation experiments on corresponding parameters.

**Weaknesses:**

Should authors explain in more detail how persistent homology is calculated? I think the part about how it is integrated into the diffusion model and how the loss between the two sets of P and Q defined is well explained, but the specific calculation of persistent homology seems a bit lacking in details.

The persistence diagram could be more explicit, for example by using different point sizes and colors to show which P sets are more important.

**Questions:**

In Figure 4, the lower part of \alpha_3 seems to be cut off by a straight line? Is it because it is a cube? If so, why doesn't it appear on the other three boundaries?

Although I know this is not the task of this paper, it would be nice if higher-dimensional topological control, such as 3D, could be discussed as future work.

---

> ### Author Response · Authors · 2024-11-22
> **Response to Reviewer mfGQ**
>
> We thank the reviewer for the encouraging and insightful comments. Please find our responses to specific queries below.
>
> **Q1: The lower part of $\alpha_3$ seems to be cut off by a straight line?**
>
> **A1:** The straight line is an intermediate effect visible during the diffusion process. Specifically, in this timestep, our $\mathcal{L}_{\text{denoise}}$ loss is actively preventing the merging of $\alpha_3$ and $\alpha_2$ components by enforcing lower intensity values at the pixels between these structures. This creates the appearance of a straight 'cut-off' line below $\alpha_3$. This effect is prominent between $\alpha_3$ and $\alpha_2$ components due to their proximity and the strength of the topological constraint in this region. It is not observed below $\alpha_1$, where there is no risk of components merging. This distinction is clearer in the $\mathcal{S}(0.21)$ image, showing how the topology-enforcing mechanism maintains separation between components.
>
> **Q2: Discuss higher-dimensional topological control, such as 3D, as future work.**
>
> **A2:** Thank you for the suggestion. Our current framework using persistent homology can theoretically be extended to higher dimensions, as persistence diagrams can capture topological features in arbitrary dimensions. For 3D applications, we can control not just connected components (0-dim) and holes (1-dim), but also voids (2-dim). As future work, we are currently investigating 3D point clouds and volumetric medical imaging data, where maintaining topology is important in generating realistic synthetic data. We have updated the same in Appendix N (page 27).
>
> **Q3: Explain in more detail how persistent homology is calculated; the persistence diagram could be more explicit.**
>
> **A3:** We would be happy to expand on the persistence homology and diagram computation. As mentioned in the implementation details, we use the Cubical Ripser library [1] for the homology computation. Their manuscript has details about how they optimized the implementation. For 0-dim, tracking the birth (creation) and death (merging) of connected components is done via the union-find algorithm. For 1-dim and higher features (e.g., loops and voids), a sparse coboundary matrix is conducted that encodes relationships between pixels. This matrix is reduced to upper-triangular form using column operations, optimizing the identification of the birth and death of cycles.
>
> In Appendix I (page 21), we now provide a detailed walkthrough example of how the 0-dim persistent diagram is computed using the union-find operations. Fig. 13 (Appendix I, page 22) consolidates the process.
>
> Thank you very much for your review. We hope we were able to clarify your comments, and we would be happy to discuss further!
>
> [1] Shizuo Kaji, Takeki Sudo, and Kazushi Ahara. Cubical ripser: Software for computing persistent homology of image and volume data. arXiv preprint arXiv:2005.12692, 2020.

---

> > ### Comment · Reviewer_mfGQ · 2024-11-25
> >
> > Thanks for the authors' response; as I have already given a high score, I will maintain my rating.

---

> > > ### Author Response · Authors · 2024-11-26
> > > **Thank you**
> > >
> > > Thank you very much for your response and recommendation. We are glad we were able to address your comments.
> > >
> > > Sincerely, Authors#3999

---

### Official Review · Reviewer_veux · 2024-11-03

**Soundness:** 3
**Presentation:** 3
**Contribution:** 3
**Rating:** 6
**Confidence:** 3

**Summary:**

This paper proposes a generative method for images that can be constrained in its resulting topology - number of objects, as well as the connectivity between objects. The core idea is to use persistent homology which essentially constructs, for each image produced in a diffusion process, a 3D embedding is created by marking all pixels that exceed each z>0 - this enables the authors to produce a topological signature for the image (for different z's) which enables to ignore noisy structures and evaluate the topological invariants of the resulting clean image. The authors then devise a simple loss that uses this information and regularize the neural network to respect the topological constraints.

**Strengths:**

I am not an expert in image generation, but I like the idea of adding topological constraints to image generation. It could be very interesting to be able to accurately control resulting images' topology.  I found the approach to be rather straightforward and thus easy to understand and implement. Considering I could not find a prior work that aims to achieve the same result, I find this approach novel.

**Weaknesses:**

Again, I am not an expert in image generation, however it seems to me that in the end, the paper focuses mainly on number of objects and less so on more important topological invariants such as connectivity. There are many interesting tests one could cook up, e.g., choose Betti numbers that force a very specific topology (double annulus) and show the method can achieve that. These more interesting topological invariants are also not directly measured in the quantitative evaluations, as far as I understand. I am also uncertain how far can this type of topological control be effective for practical uses, beyond, again, the trivial case of controlling number of objects.

**Questions:**

- Empirically (not in theory), can your code generate more complex specific shapes by choosing the right topological invariants as input?

---

> ### Author Response · Authors · 2024-11-22
> **Response to Reviewer veux**
>
> We thank the reviewer for the encouraging and insightful comments. Please find our responses to specific queries below.
>
> **Q1: Empirically (not in theory), can your code generate more complex specific shapes like double annulus?**
>
> **A1:** By combining multiple topological constraints, our method can provide more sophisticated control. To show this, we conduct experiments using both 0-dim and 1-dim topological constraints simultaneously. We represent these joint constraints as pairs $(a,b)$, where $a$ specifies the number of connected components (0-dim) and $b$ specifies the number of holes (1-dim). The performance from Tab. 9 (Appendix K, page 24), or, the second table in the Global Response, shows that TDN can handle joint constraints.
>
> A double annulus consists of concentric circles, with $(a,b) = (2,2)$. Our loss, based on homology, cannot distinguish whether two loops are positioned concentric or not, however, it can generate structures that are homotopy equivalent to the double annulus. As shown in Fig. 15 (Appendix K, page 24), by sampling for the $(2,2)$ condition, TDN can generate nested circles (visually the closest constraint to a double annulus) and other homotopy equivalent structures. This experiment demonstrates that our method can handle complex topological specifications through the combination of constraints, though geometric relationships (like equidistance and concentricity) remain an interesting direction for future work.
>
> **Q2: Topological control for practical uses, beyond the trivial case of controlling the number of objects.**
>
> **A2:** While counting may seem trivial for humans, it is already a challenging task for diffusion models, and in general generative models. As shown in Tab. 1 and Fig. 6, even state-of-the-art models like DALL·E 3 and Stable Diffusion fail to reliably generate a specified number of objects. Our novel approach provides control in both counting (0-dim topology), and number of holes (1-dim topology). This is a significant first step towards semantic control of generative models, moving beyond qualitative attributes such as style and texture. As future work, we are currently investigating 3D point clouds and volumetric medical imaging data, where preserving topology is important in generating realistic data.
>
> Thank you very much for your review. We hope we were able to clarify your comments, and we would be happy to discuss further!

---

> > ### Author Response · Authors · 2024-11-26
> > **Regarding the Rebuttal**
> >
> > We thank the reviewer for their feedback, and hope our responses have effectively addressed all of the reviewer’s comments. We would be grateful to know if there are any additional questions that we can clarify; we would be happy to discuss further. Thank you!

---

> > ### Comment · Reviewer_veux · 2024-11-26
> >
> > Thank you for your response.
> > First, I note that usually the term k-annulus implies that k denotes the number of holes, similarly as a k-torus implies that k denotes the number of handles. Hence, the circles need not be cocentric, as far as I know.
> >
> > Second, while defining number of disconnected components (what you refer to as "counting") as a topological invariant is technically correct, I don't understand why using heavy machinery and notions such as homology are strictly necessary for such a relatively trivial invariant. As I see it, if these tools indeed provide capabilities for exact control on topology, I would like to see results such as an image of 5 objects, with 3 holes in total.
> >
> > I again note that since I am not an expert in 2D image generation, I refrain from neither lowering nor raising my score - this is my best evaluation given my level of acquaintance with this area of research.

---

> > > ### Author Response · Authors · 2024-11-28
> > > **Response to Reviewer veux**
> > >
> > > We thank the reviewer for their response, and address the comments as below.
> > >
> > > **Q1: Results such as an image of 5 objects, with 3 holes in total.**
> > >
> > > **A1:** We interpret this as you would like to see real images using the joint topological constraints. In the previous response, we used joint topological conditions (0-dim connected components and 1-dim holes simultaneously) to generate masks using our proposed TDN. We now provide the corresponding images rendered by ControlNet. The rendered images contain a mix of connected components (burgers / eggs / coins) and holes (donuts / rings). Please see Fig. 14 (Supplementary Material pdf, Appendix K, page 24).
> > >
> > > **Q2: Why using heavy machinery and notions such as homology is strictly necessary for such a relatively trivial invariant.**
> > >
> > > **A2:** Counting may seem straightforward to humans, however, teaching a diffusion model to generate a mask with the correct count is much more challenging. The key difficulty lies in guiding the model toward the correct topology when the generated mask is initially incorrect. To address this, we introduce a persistent-homology-based loss function. Persistent homology not only computes topology, but also measures them from noisy observations, quantifying the distance from the desired topological constraints. This loss function effectively guides the reverse denoising diffusion process to generate masks that preserve topology. Consequently, our TDN preserves topology with high fidelity, as demonstrated empirically, whereas previous diffusion models have consistently failed.
> > >
> > > We hope we have adequately addressed the reviewer’s questions and concerns. We sincerely appreciate the valuable feedback and discussion, which have greatly enriched our work. We will incorporate these discussions into the final version of the paper. Thank you.

---

### Official Review · Reviewer_GDUJ · 2024-11-04

**Soundness:** 3
**Presentation:** 3
**Contribution:** 2
**Rating:** 3
**Confidence:** 5

**Summary:**

This paper proposes a procedure for conditioning text-to-image diffusion models on topological constraints. In particular, an auxiliary diffusion model is trained that generates binary segmentation masks that conform to a specified zeroth and first homology ranks (i.e., the number of connected components and holes). This is done both by providing the Betti numbers as input conditions to the network and by imposing a loss on the homology of the predicted image. Because the predicted images at early timesteps are very noisy, an approximation of the denoised image is computed, and the persistence module of that image is what is used in the loss objective.

**Strengths:**

Maintaining desired compositionality is a key challenge for diffusion models, and getting a model to output the desired quantity of discrete objects is often difficult. Framing this problem through the lens of persistent homology makes a lot of sense and ties nicely to the difficulty of dealing with noisy samples at intermediate timesteps. Additionally, being able to condition on first homology is also a nice feature, albeit it seems a bit more contrived (and indeed the main demonstrated use case is in generating aerial road images).

**Weaknesses:**

My main concerns with this paper have to do with the overall novelty as well as the overall justification for the approach.

While this paper is indeed the first to consider diffusion-based image generation through the lens of TDA and persistent homology, the actual machinery proposed to accomplish this is very similar to prior work that combines TDA with deep learning --- a key insight is differentiating through the persistence computation, but this is not unique to the diffusion setting. I do agree that the observation that the a persistence module can well represent the homology of a noisy $x_0$ approximation is interesting, but this brings me to my second concern, which is that I am not convinced by the results in this paper that training a diffusion model is critical to the success of this method.

One of the main reasons for the success of image diffusion models is their stability and performance in generating high-quality complex natural images. The masks produced by the diffusion model in this paper, however, are quire primitive and lack detail or high-frequency detail. I wonder if a much simpler generative model, e.g., a VAE, could be just as effective at producing these masks, that could then equally serve as input to a ControlNet. Moreover, I am not convinced that an ML model is necessary to generate these mattes in the first place. Given that the only conditions that the model accepts are the Betti numbers and an object class (nothing more about the actual layout can be specified), it seems to me that such masks could be procedurally generated to obey the desired topology. In fact, looking at Figure 6, it appears that the object class condition is superfluous for the mask model, given that it also appears in the ControlNet prompt. If we look, e.g., at the birds (bottom right), the contour of the second bird from the right is completely different from the input mask. This suggests that the ControlNet largely ignores any higher-frequency context in the mattes.

**Questions:**

It would be very helpful if the authors could respond to my two points noted above---the second one in particular. At the very least, some simple ablation studies can be conducted to help justify the overall algorithm. For instance, what if the conditioning masks simply contain circles or other basic shapes for the desired objects (even in the case of animals)? Would the ControlNet still understand the object type from the text prompt and use the simplified mask as the cardinality/topology conditioning?

---

> ### Author Response · Authors · 2024-11-22
> **Response to Reviewer GDUJ [1/2]**
>
> We thank the reviewer for the constructive feedback. The questions help us improve the justification of our work. Please find our responses to specific queries below.
>
> **Q1: The masks produced by the diffusion model in this paper lack detail. Hence, are diffusion models critical to the success of this method?**
>
> **A1:** While the masks generated by our method TDN may appear simplistic, they effectively capture the overall structure of the objects, and are in fact, not trivial to generate. Furthermore, the details of the TDN-generated masks are necessary and sufficient for generating realistic and semantically correct images.
>
> Despite being binary images, these masks are difficult to generate due to variations in the number of objects, their shapes, sizes, and unconstrained spatial locations (e.g., not fixed to the center of the image). Similarly, the diverse orientation and connectivity patterns make the 1-dim datasets (road networks, neuron membranes) challenging. This is evidenced by the results below (in Q2): we show that a conditional VAE cannot learn to generate satisfying masks, even with the topology-preserving loss. Diffusion models are thus critical to the success of our method as they effectively capture spatial arrangements and object shapes (in 0-dim), and connectivity patterns (in 1-dim).
>
> Moreover, the details in the masks generated by TDN are necessary for generating realistic images. As suggested, in Q3, we report results using basic shapes (e.g.: circles and squares) as masks to guide ControlNet. These simplified masks, although have the correct topology, result in poor-quality images. In contrast, TDN-generated masks provide much more detailed geometric, spatial and topological control which are critical to the generation of realistic images.
>
> Finally, we stress that TDN-generated masks are sufficient to generate high quality images, thanks to the power of ControlNet. This is validated by our quantitative results (see Tab. 8 (Appendix J, page 23), or, the first table in the Global Response). We observe similar visual quality between TDN-generated masks and ground truth (GT) masks (low FID [Mask] scores), as well as similar visual quality between images generated using these masks (comparable FID [Image] score).
>
> Overall, our masks strike a balance between being oversimplified and unnecessarily complex, and diffusion models are capable of filling in the finer textural details based on these masks.
>
> **Q2: Can VAE be just as effective at generating masks to condition ControlNet?**
>
> **A2:** We address this concern by conducting experiments using a Conditional VAE (CVAE) as the generative model. Fig. 16 (Appendix L, page 25) shows that the quality of the masks generated by CVAE is severely degraded, with and without our topology-preserving loss. In the table below, CVAE also has significantly worse FID compared to our TDN-generated masks, indicating a poor visual resemblance to the true masks. CVAE also struggles to preserve topological constraints, as evidenced by the low performance on metrics like Accuracy and F1.
>
> We also attempted to add our proposed topology-preserving loss $\mathcal{L}\_{top}$ to CVAE. As shown in the table, this approach (CVAE + $\mathcal{L}\_{top}$) does slightly improve performance over the standard CVAE. However, the results remain far inferior to those of diffusion models (ADM-T and TDN). Specifically, for 0-dim datasets, it generates fragmented objects and fails to preserve overall shapes, while for 1-dim datasets, it fails to generate connected structures.
>
> This is not surprising. CVAE’s bottleneck layer compresses information, losing critical spatial details needed to enforce strict topological constraints. In contrast, diffusion models’ design of gradual denoising steps, as well as intact image resolution, ensure the preservation of details like shape, with topology being preserved because of our novel loss.

---

> > ### Author Response · Authors · 2024-11-22
> > **Response to Reviewer GDUJ [2/2]**
> >
> > | **Dataset (TopoDim)**    | **Method**              | **Accuracy ↑**           | **F1 ↑**                | **FID (Mask) ↓** |
> > |---------------------------|-------------------------|--------------------------|-------------------------|------------------|
> > | **Shapes (0-dim)**        | CVAE        | 0.3133 ± 0.1321          | 0.3078 ± 0.1443         | 2.231           |
> > |                           | CVAE + 𝓛ₜₒₚ             | 0.3816 ± 0.0919          | 0.3220 ± 0.1365         | 1.981           |
> > |                           | ADM-T                  | 0.7500 ± 0.1889          | 0.7651 ± 0.1210         | 0.092           |
> > |                           | TDN (Ours)            | **0.9478 ± 0.0420**      | **0.9488 ± 0.0370**     | **0.068**       |
> > | **COCO (0-dim)**          | CVAE        | 0.2442 ± 0.0410          | 0.2562 ± 0.0318         | 4.208           |
> > |                           | CVAE + 𝓛ₜₒₚ             | 0.3094 ± 0.0967          | 0.3213 ± 0.1165         | 4.083           |
> > |                           | ADM-T                  | 0.6685 ± 0.1485          | 0.6799 ± 0.1931         | 0.267           |
> > |                           | TDN (Ours)            | **0.8557 ± 0.0805**      | **0.8613 ± 0.0970**     | **0.222**       |
> > | **CREMI (1-dim)**         | CVAE        | 0.2785 ± 0.4296          | 0.2499 ± 0.1391         | 5.971           |
> > |                           | CVAE + 𝓛ₜₒₚ             | 0.3267 ± 0.2834          | 0.3176 ± 0.1447         | 5.751           |
> > |                           | ADM-T                  | 0.5357 ± 0.1879          | 0.4881 ± 0.1571         | 0.518           |
> > |                           | TDN (Ours)            | **0.7785 ± 0.1901**      | **0.7959 ± 0.1659**     | **0.467**       |
> >
> > **Q3: Would ControlNet interpret object types from the text prompt if the conditioning masks only contain basic shapes like circles/squares, using them for cardinality/topology conditioning?**
> >
> > **A3:** To address this, we conduct experiments using simple shapes (circles and squares) as conditioning masks for ControlNet. Our results, in Fig. 17 (Appendix M, page 26) and the table below show that such masks, even with correct topology, fail to provide sufficient guidance to ControlNet. First, from the qualitative results in Fig. 17, we see that such conditions tend to introduce visual artifacts in the generated images. The images are often fragmented, with objects isolated within assigned areas, leading to a divided and visually disjointed image. Second, the basic shapes also fail to constrain the number of objects within them, often resulting in 0 objects or multiple objects per shape (see the ‘cats’ row in Fig. 17). Finally, for large animals like zebras, ControNet struggles to complete the image beyond the boundary of the shape—while the zebra texture is present within the shape, the overall image is not as desired. Similarly, for even larger animals like giraffes, the discrepancy between the basic shape and the natural object proportions leads to incomplete or distorted generations.
> >
> > To some extent, this approach is similar to the AR method (Tab. 1, Fig. 6), which uses rectangular bounding boxes with an attention step and faces similar limitations. In contrast, TDN masks provide tighter spatial and structural control while maintaining sufficient detail to guide the network effectively. This results in correct object counts and better visual quality, as evidenced by superior FID scores (Tab. 8) and count metrics (Tab. 1). Our results highlight that relying solely on a text prompt and simplified masks is insufficient for the model to generate visually coherent images. Instead, meaningful masks like those generated by TDN are essential for ensuring reliable control over quality, cardinality, and topology in generative models.
> >
> > | **Method**      | **Accuracy ↑**           |      | **Precision ↑**         |      | **F1 ↑**                |      | **FID (Image) ↓** |
> > |----------------------------------------|--------------------------|------|-------------------------|------|-------------------------|------|------------------|
> > | Basic circles mask + ControlNet                    | 0.2972 ± 0.3592          |      | 0.3194 ± 0.3046         |      | 0.3079 ± 0.2208         |      | 49.47           |
> > | Basic squares mask + ControlNet                     | 0.3123 ± 0.2114          |      | 0.3459 ± 0.2209         |      | 0.3282 ± 0.2618         |      | 47.16           |
> > | AR (Bounding box w/ attention)         | 0.6379 ± 0.2062          |      | 0.7360 ± 0.1658         |      | 0.6611 ± 0.1851         |      | 35.05           |
> > | TDN (Ours)                             | **0.8557 ± 0.0805**      |      | **0.8670 ± 0.0636**     |      | **0.8613 ± 0.0970**     |      | **21.28**       |
> >
> >
> > Thank you very much for your review. We hope we were able to clarify your questions regarding the use of diffusion models, and we would be happy to discuss further!

---

> > > ### Author Response · Authors · 2024-11-26
> > > **Regarding the Rebuttal**
> > >
> > > We thank the reviewer for their feedback, and hope our responses have effectively addressed all of the reviewer’s comments. We would be grateful to know if there are any additional questions that we can clarify; we would be happy to discuss further. Thank you!

---

> > > > ### Comment · Reviewer_GDUJ · 2024-11-26
> > > > **Response**
> > > >
> > > > Thank you to the authors for the response and additional experiments. While some of my concerns have been addressed, I am a bit puzzled by the quality of results in the new Figure 17. As a point of reference, the paper "Iterative Object Count Optimization for Text-to-image Diffusion Models" [Zafar et al. 2024]  Figure 4 has an almost identical experiment and achieves much more reasonable results. Note, that I am not considering this paper when evaluating your submission, as it is concurrent work, but I do wonder why there is such a discrepancy in a baseline experiment.

---

> > > > > ### Author Response · Authors · 2024-11-28
> > > > > **Response to Reviewer GDUJ**
> > > > >
> > > > > We thank the reviewer for their response, and would like to address the difference in images generated by us, and those in the suggested reference [1]. In our implementation details, we mention that we use the popular SD1.5 as the ControlNet backbone, while [1] uses the latest SDXL-Turbo [2] backbone whose checkpoint was released in July 2024 [3]. SDXL-Turbo has superior image generation quality as presented in [2]. Hence, we now render images of our TDN-generated masks using the new backbone and provide the results in Fig. 18, Fig. 19, and Fig. 20 (Supplementary Material pdf, Appendix M.1, pages 28-30).
> > > > >
> > > > > Using the new backbone, we find that while the artifacts using basic shapes are no longer present, ControlNet still largely ignores the basic shapes’ spatial arrangement and cardinality. We hypothesize that due to the gap between the basic shape and the complex real object contour (like animals), the powerful diffusion model opts to drop the conditional mask and completely rely on its inherent knowledge. In contrast, using masks from TDN results in correct counts, with the diffusion model strictly following the masks’ spatial arrangement and topology.
> > > > >
> > > > > Hence, we find that even with the latest backbone, relying solely on a text prompt and simplified masks is insufficient for the model to generate semantically correct images. Instead, meaningful masks like those generated by TDN are essential for ensuring reliable control over cardinality and topology in generative models.
> > > > >
> > > > > We hope that we were able to clarify the difference in results, and justify the importance of using TDN masks to control topology. We sincerely appreciate the valuable feedback and discussion, as it has strengthened the motivation of our work. We would be grateful to know if there are any additional questions that we can clarify; we would be happy to discuss further. Thank you.
> > > > >
> > > > > [1] Zafar, Oz, Lior Wolf, and Idan Schwartz. "Iterative object count optimization for text-to-image diffusion models." arXiv preprint arXiv:2408.11721 (2024).
> > > > >
> > > > > [2] Sauer, Axel, et al. "Adversarial diffusion distillation." European Conference on Computer Vision. Springer, Cham, 2025.
> > > > >
> > > > > [3] https://huggingface.co/stabilityai/sdxl-turbo

---

> ### Author Response · Authors · 2024-12-01
> **Following-up**
>
> We sincerely thank reviewer GDUJ for their time and effort in reviewing and discussing our paper. We hope our previous response has addressed all of the reviewer’s comments. As the rebuttal deadline approaches, we would be grateful to know if there are any additional questions that we can clarify; we would be happy to discuss further. Thank you.
>
> Sincerely, Authors#3999

---

> > ### Author Response · Authors · 2024-12-03
> > **Following-up 2**
> >
> > We sincerely thank reviewer GDUJ for their time and effort in reviewing and discussing our paper. We hope our previous response has addressed all of the reviewer’s comments. As the rebuttal deadline is less than a day away, we would be grateful to know if there are any additional questions that we can clarify; we look forward to the reviewer's response. Thank you.
> >
> > Sincerely, Authors#3999

---

### Official Review · Reviewer_1o9Y · 2024-11-04

**Soundness:** 3
**Presentation:** 3
**Contribution:** 3
**Rating:** 8
**Confidence:** 3

**Summary:**

The paper proposes TopoDiffusionNet, a method to control some topological properties of 2D image generation, such as the number of distinct objects and separated regions.

TopoDiffusionNet is trained to take as input a condition $c$, representing the number of the structures, and to output a mask with exactly $c$ separate objects or regions. The network is optimized with a loss based on persistent homology; after computing the persistent diagram, the features are separated into "preserve" (the $c$ features with higher persistence), and "noise" (all the others). The network is trained to promote the first while suppressing the others. The generated mask can be used as a condition of a 2D generative ControlNet.

The method is tested on four datasets (Shapes, COCO, CREMI, Google Maps), showing significantly more control than previous methods.

**Strengths:**

The posed problem is interesting, and it is relevant for a wide community. Defining new ways to control generation and especially incorporating this knowledge into the diffusion process itself provide interesting insights. The paper is also informative and well-organized: The text is well-written, and it is pretty straightforward. The proposed methodology is clear, and the figures are quite insightful. Experiments consider diverse datasets and control results are convincing.

**Weaknesses:**

My main concern is that the paper does not offer much insight into the trade-off between control and quality. It is pretty natural to wonder how the control impacts the quality of the generation compared with other methods. The qualitative examples shown seem ok, but it is also evident that the quality is quite degraded compared to alternatives -- this might be a cause of the generative 2D ControlNet itself or that the control does not incorporate enough domain knowledge and provide a mask difficult to respect (e.g., the mask of cats and birds do not look realistic, and I guess it requires quite some work for the ControlNet generation to compensate) . The only FID score is reported in an ablation study on the Shape dataset, which is not particularly insightful. I think reporting (at least) the same metric for other datasets and comparing it with other methods would be important to assess how much the provided approach destroys the generative capability of the generative backbone.

**Questions:**

Please refer to the weaknesses of the main points to clarify.

Some minor curiosities are:

1) Would it be possible to combine 0-dim and 1-dim constraints?
2) Does the computation of persistent homology have an impact on the computational performance of the method?
3) While the method provides control only on the number $c$, is it possible to tune homology parameters to promote also bigger or smaller components?

---

> ### Author Response · Authors · 2024-11-22
> **Response to Reviewer 1o9Y**
>
> We thank the reviewer for the encouraging and insightful comments. Please find our responses to specific queries below.
>
> **Q1: Insight into the trade-off between control and quality. Reporting FID on all datasets.**
>
> **A1:** This is a valid question. Our method TDN achieves improved topological control without sacrificing the overall visual quality of the generated images. We demonstrate this by providing FID scores for all the datasets in Tab. 8 (Appendix J, page 23), or, the first table in the Global Response. Recall that there are two generation outputs: masks, and images conditioned on the masks. Our method TDN focuses on generating masks with correct topology, and delegates to ControlNet to generate images conditioned on the masks. We report both the FID of the mask (FID [Mask]) and the FID of the images generated by ControlNet when using these masks (FID [Image]).
>
> Our analysis shows that across all datasets, the masks generated by TDN closely resemble the ground truth (GT) masks, in both visual quality (low FID [Mask]) and topology (Tab. 1). We also observe high visual quality in images generated using the masks from TDN (low FID [Image] compared to baselines).
>
> Additionally, to evaluate the impact on final image quality, we compare the FID [Image] metric between two scenarios: ControlNet using GT masks (ControlNet + GT) versus ControlNet using TDN-generated masks. The results demonstrate comparable FID scores, indicating that using TDN-generated masks as conditioning does not degrade the image quality. In other words, TDN-generated masks are as good as GT masks for generating real images.
>
>
> **Q2: Would it be possible to combine 0-dim and 1-dim constraints?**
>
> **A2:** This is an interesting set of constraints. In our work, we focused on using either 0-dim (number of objects) or 1-dim (number of holes) constraints individually, depending on the property of the dataset. However, as suggested, we conduct experiments on synthetic data using both 0-dim and 1-dim constraints simultaneously. The constraint is of the form $(a,b)$ where $a$ denotes 0-dim while $b$ denotes 1-dim. Sample images generated by TDN are shown in Fig. 14 (Appendix K, page 24). We report quantitative results in Tab. 9 (Appendix K, page 24), or, the second table in the Global Response.
>
> We find that TDN is capable of handling these joint constraints. While the performance is slightly lower compared to using each constraint individually, TDN still significantly outperforms ADM-T across all metrics. This shows that persistent homology (PH) can enhance the performance of the base diffusion model, even under multi-constraint scenarios. This finding opens up interesting possibilities for future work in handling even richer combinations of topological constraints.
>
>
> **Q3: Computational performance of the method.**
>
> **A3:** We use persistent homology only in the training phase to preserve the desired structures and penalize the extraneous ones. During inference, persistent homology is not used, hence there is no additional computational overhead or impact on inference time.
>
> **Q4: Is it possible to tune homology parameters to promote also bigger or smaller components?**
>
> **A4:** Good question. Yes, indeed, it is possible to tune parameters to control the size of the components. Given a mask, by flipping the foreground and background, applying a distance transform, and computing the 1-dim persistence diagram, the death time would reflect the size or radius of the object. This works because with the distance transform, the local maximum value corresponds to the farthest distance from the boundary, effectively encoding the object’s radius. Encoding the size constraint in this manner is interesting but could provide relatively limited flexibility. Much richer geometric prior can be injected using other topological tools such as *persistent homology transform* [1]; this will be an exciting direction to explore in future works.
>
> Thank you very much for your review. We hope we were able to clarify your comments, and we would be happy to discuss further!
>
> [1] Turner, Katharine, Sayan Mukherjee, and Doug M. Boyer. "Persistent homology transform for modeling shapes and surfaces." Information and Inference: A Journal of the IMA 3, no. 4 (2014): 310-344.

---

> > ### Comment · Reviewer_1o9Y · 2024-11-22
> > **Post rebuttal**
> >
> > I thank the authors for their efforts in answering my questions. The further experimental evidence addresses my curiosity well, showing the flexibility of the proposed approach, its wide applicability, and possible further extensions.
> >
> > As a minor element, I would probably include some images generated using the 0-dim and 1-dim jointly, to show how the generative backbone supports this detailed control.
> >
> > As of now, I have retained my initial positive consideration of the paper, and I am looking forward to hearing other reviewers' opinions.

---

> > > ### Author Response · Authors · 2024-11-22
> > > **Thank you for your response**
> > >
> > > Thank you very much for your response! We are glad we were able to address your comments.
> > >
> > > Regarding images generated using 0-dim and 1-dim joint constraints, we have included sample images generated by TDN in the updated pdf.  We include them in figures Fig. 14 and Fig. 15 (Appendix K, page 24).
> > >
> > > Sincerely, Authors#3999

---

> > > > ### Comment · Reviewer_1o9Y · 2024-11-25
> > > > **Thx**
> > > >
> > > > Thank you -- I have seen those Figures, but I was mainly referring to actual images obtained from such control, to demonstrate that the whole pipeline is able to handle this (i.e., similar to figure 17). But it is just a minor comment, intended for completeness. I do not have further questions.
> > > >
> > > > Best.

---

> > > > > ### Author Response · Authors · 2024-11-28
> > > > > **Thank you**
> > > > >
> > > > > We thank the reviewer for the clarification. Earlier, we provided visual results of our TDN-generated masks using 0-dim and 1-dim joint constraints. We now provide the corresponding images rendered by ControlNet. The rendered images contain a mix of connected components (burgers / eggs / coins) and holes (donuts / rings). Please see Fig. 14 (Supplementary Material pdf, Appendix K, page 24).
> > > > >
> > > > > We sincerely thank the reviewer for their valuable feedback and recommendation.

---

### Author Response · Authors · 2024-11-22
**Global Response To All Reviewers**

We thank all the reviewers for their time and insightful feedback. We are encouraged that the reviewers found our proposed approach well-motivated and relevant (1o9Y, GDUJ, veux, mfGQ), with sufficient experiments demonstrating its effectiveness (1o9Y, mfGQ). The reviewers also appreciate the clear writing and presentation of the paper (1o9Y, mfGQ).

We address each reviewer's specific questions individually. In this global response, we present the requested experimental results; we provide a detailed discussion of these results within the corresponding individual responses. We have also updated the manuscript with these findings.

**1. Comparison of FID (1o9Y,GDUJ)**. For ADM-T and TDN, we provide the FID of the mask [FID (Mask)] and the FID of the images generated by ControlNet when using these masks [FID (Image)]. For the baselines, we provide their FID (Image). For Shapes and Google Maps datasets, we cannot report FID (Image) as there is no ground truth image collection to compare to. For COCO and CREMI, we report FID (Image) against the respective dataset images. 'ControlNet + GT’ indicates that ControlNet is using the GT masks as the condition. 'ControlNet + FT’ indicates that it has been fine-tuned on the CREMI dataset to generate similarly textured images. **Bold** denotes the best results, while *italics* denotes the second best. Note that ControlNet + GT has been included as the upper bound performance; realistically, it cannot be used during inference as the GT masks are not available. We have included the same table as **Tab. 8** in Appendix J (page 23).

| **Dataset**       | **Method**           | **FID (Mask) ↓** | **FID (Image) ↓** |
|--------------------|----------------------|------------------|-------------------|
| **Shapes**        | ADM-T               | 0.092            | -                 |
|                   | TDN (Ours)          | **0.068**        | -                 |
| **COCO (Animals)**| Stable Diffusion    | -                | 29.41             |
|                   | DALL·E 3            | -                | **17.49**         |
|                   | AR                  | -                | 35.05             |
|                   | ADM-T               | 0.267            | 21.72             |
|                   | TDN (Ours)          | **0.222**        | *21.28*           |
|                   | ControlNet + GT     | -                | 20.94             |
| **Google Maps**   | ADM-T               | 0.198            | -                 |
|                   | TDN (Ours)          | **0.156**        | -                 |
| **CREMI**         | Stable Diffusion    | -                | 48.18             |
|                   | DALL·E 3            | -                | 54.72             |
|                   | AR                  | -                | 69.86             |
|                   | ADM-T               | 0.518            | 3.322             |
|                   | TDN (Ours)          | **0.467**        | **3.286**         |
|                   | ControlNet + GT + FT| -                | 3.126             |


**2. Complex topological constraints (1o9Y,veux)**. We conduct experiments on synthetic data using both 0-dim and 1-dim constraints simultaneously.  The table shows the quantitative performance of using these joint constraints. The FID (Mask) is reported once for each method. Best results are in **bold**. We have included the same table as **Tab. 9** in Appendix K (page 24).

| **TopoDim** |      | **Method**    |      | **Accuracy ↑**           |      | **Precision ↑**         |      | **F1 ↑**                |      | **FID (Mask) ↓** |
|-------------|------|---------------|------|--------------------------|------|-------------------------|------|-------------------------|------|------------------|
| **0-dim**   |      | ADM-T         |      | 0.7383 ± 0.1305          |      | 0.7997 ± 0.1268         |      | 0.7677 ± 0.1229         |      | 0.1279           |
|             |      | TDN (Ours)    |      | **0.9183 ± 0.0731**      |      | **0.9338 ± 0.0993**     |      | **0.9261 ± 0.0906**     |      | **0.0982**       |
| **1-dim**   |      | ADM-T         |      | 0.7616 ± 0.0905          |      | 0.7892 ± 0.1129         |      | 0.7752 ± 0.1082         |      | -                |
|             |      | TDN (Ours)    |      | **0.9233 ± 0.0705**      |      | **0.9492 ± 0.0913**     |      | **0.9360 ± 0.0720**     |      | -                |

---

### Meta-Review · Area_Chair_rZ4H · 2024-12-14

**Metareview:**

The authors propose a method to introduce topological constraints into the generation process of image diffusion models. The method splits the generation process in two parts. First, masks are generated with topology guidance based on persistence diagrams. Then, the image is generated via ControlNet conditioned on the generated mask. The paper clearly shows the benefits of this design in the experiments, when generating instances with specified topology.

The reviews for this paper were 3 vs. 1 leaning positive. The positively-leaning reviewers found the paper to be interesting and sufficiently evaluated. The negative leaning reviewer questions the necessity of including topology in the generation process and was concerned about result quality.

In this diverging situation I took a closer look at the paper and decided to follow the positive evaluation. I found this paper to be interesting and insightful, providing a potential solution to a problem that we see in many generative image diffusion models. The experimental evaluation convinces me that the presented approach leads to clear benefits. The concepts introduced might also spark additional research in other domains.

**Additional Comments On Reviewer Discussion:**

The scores were set after the first round with no changes afterwards. The authors tried to address the concerns of reviewer GDUJ who was not convinced by the answers. During the reviewer discussion, the reviewers remained with their position.

---

### Decision · Program_Chairs · 2025-01-22

Accept (Poster)